# MULTI-OBJECTIVE OPTIMIZATION BY LEARNING SPACE PARTITIONS

**Yiyang Zhao**
Worcester Polytechnic Institute

**Linnan Wang**
Brown University

**Kevin Yang**
UC Berkeley

**Tianjun Zhang**
UC Berkeley

**Tian Guo**
Worcester Polytechnic Institute

**Yuandong Tian**
Facebook AI Research

## ABSTRACT

In contrast to single-objective optimization (SOO), multi-objective optimization (MOO) requires an optimizer to find the Pareto frontier, a subset of feasible solutions that are not dominated by other feasible solutions. In this paper, we propose LaMOO, a novel multi-objective optimizer that learns a model from observed samples to partition the search space and then focus on promising regions that are likely to contain a subset of the Pareto frontier. The partitioning is based on the *dominance number*, which measures "how close" a data point is to the Pareto frontier among existing samples. To account for possible partition errors due to limited samples and model mismatch, we leverage Monte Carlo Tree Search (MCTS) to exploit promising regions while exploring suboptimal regions that may turn out to contain good solutions later. Theoretically, we prove the efficacy of learning space partitioning via LaMOO under certain assumptions. Empirically, on the HyperVolume (HV) benchmark, a popular MOO metric, LaMOO substantially outperforms strong baselines on multiple real-world MOO tasks, by up to 225% in sample efficiency for neural architecture search on Nasbench201, and up to 10% for molecular design.

## 1 INTRODUCTION

Multi-objective optimization (MOO) has been extensively used in many practical scenarios involving trade-offs between multiple objectives. For example, in automobile design (Chang, 2015), we must maximize the performance of the engine while simultaneously minimizing emissions and fuel consumption. In finance (Gunantara, 2018), one prefers a portfolio that maximizes the expected return while minimizing risk.

Mathematically, in MOO we optimize $M$ objectives $\mathbf{f}(\mathbf{x}) = [f_1(\mathbf{x}), f_2(\mathbf{x}), \ldots, f_M(\mathbf{x})] \in \mathbb{R}^M$:

$$\min \quad f_1(\mathbf{x}), f_2(\mathbf{x}), ..., f_M(\mathbf{x}) \tag{1}$$
$$\text{s.t.} \quad \mathbf{x} \in \Omega$$

While we could set arbitrary weights for each objective to turn it into a single-objective optimization (SOO) problem, modern MOO methods aim to find the problem's entire *Pareto frontier*: the set of solutions that are not *dominated* by any other feasible solutions[1] (see Fig. 1 for illustration). The Pareto frontier yields a global picture of optimal solution structures rather than focusing on one specific weighted combination of objectives.

As a result, MOO is fundamentally different from SOO. Instead of focusing on a single optimal solution, a strong MOO optimizer should cover the search space broadly to explore the Pareto frontier. Popular quality indicators in MOO, such as hypervolume (HV), capture this aspect by computing the volume of the currently estimated frontier. Specifically, given a reference point $R \in \mathbb{R}^M$, as shown in Fig. 1(a), the *hypervolume* of a finite approximate Pareto set $\mathcal{P}$ is the M-dimensional

---

[1]Here we define *dominance* $\mathbf{y} \prec_{\mathbf{f}} \mathbf{x}$ as $f_i(\mathbf{x}) \leq f_i(\mathbf{y})$ for all functions $f_i$, and exists at least one $i$ s.t. $f_i(\mathbf{x}) < f_i(\mathbf{y})$, $1 \leq i \leq M$. That is, solution $\mathbf{x}$ is always better than solution $\mathbf{y}$, regardless of how the $M$ objectives are weighted.

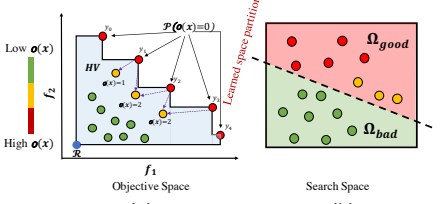

| MOO methods | Sampling Method | Objectives>3 |
|---|---|---|
| MOEA/D (Zhang & Li, 2007) | | × |
| CMA-ES (Igel et al., 2007a) | Evolution | × |
| NSGA-II (Deb et al., 2002a) | | × |
| NAGA-III (Deb & Jain, 2014) | | √ |
| PAREGO (Knowles, 2006) | Bayesian optimization | √ |
| qEHVI (Daulton et al., 2020) | | √ |
| **LaMOO** (our approach) | Space partition | √ |

Figure 1: **Left**: A basic setting in Multi-objective Optimization (MOO), optimizing $M = 2$ objectives in Eqn. 1. (a) depicts the objective space $(f_1, f_2)$ and (b) shows the search space $\mathbf{x} \in \Omega$. In (a), $P$ denotes the Pareto frontier, $R$ is the reference point, the hypervolume $HV$ is the space of the shaded area, and $o(\mathbf{x})$ are the dominance numbers. In (b), once a few samples are collected within $\Omega$, LaMOO *learns to partition* the search space $\Omega$ into sub-regions (i.e. $\Omega_{good}$ and $\Omega_{bad}$) according to the dominance number in objective space, and then focuses future sampling on the good regions that are close to the Pareto Frontier. This procedure can be repeated to further partition $\Omega_{good}$ and $\Omega_{bad}$. **Right**: A table shows the properties of MOO methods used in experiments.

Lebesgue measure $\lambda_M$ of the space dominated by $\mathcal{P}$ and bounded from below by $R$. That is, $HV(\mathcal{P}, R) = \lambda_M(\cup_{i=1}^{|\mathcal{P}|}[R, y_i])$, where $[R, y_i]$ denotes the hyper-rectangle bounded by reference point $R$ and $y_i$. Consequently, the optimizer must consider the diversity of solutions in addition to their optimality.

While several previous works have proposed approaches to capture this diversity-optimality trade-off (Deb et al., 2002a; Knowles, 2006; Igel et al., 2007; Deb & Jain, 2014; Daulton et al., 2020), in this paper, we take a fundamentally different route by *learning* promising candidate regions from past explored samples. Ideally, to find the Pareto frontier in as few function evaluations as possible, we want to sample heavily in the Pareto optimal set $\Omega_P$, defined as the region of input vectors that corresponds to the Pareto frontier.

One way to focus samples on $\Omega_P$ is to gradually narrow the full search space down to the subregion containing $\Omega_P$ via partitioning. For example, in the case of quadratic objective functions, $\Omega_P$ can be separated from the non-optimal set $\Omega \backslash \Omega_P$ via simple linear classifiers (see Observation 1,2). Motivated by these observations, we thus design LaMOO, a novel MOO meta-optimizer that progressively partitions regions into sub-regions and then focuses on sub-regions that are likely to contain Pareto-optimal regions, where existing solvers can help. Therefore, LaMOO is a meta-algorithm.

Unlike cutting-plane methods (Loganathan & Sherali, 1987; Hinder, 2018; Vieira & Lisboa, 2019) that leverage the (sub)-gradient of convex objectives as the cutting plane, with global optimality guarantees, LaMOO is data-driven: it leverages previous samples to build classifiers to *learn* the partition and focuses future samples in these promising regions. No analytical formula of objectives or their sub-gradients is needed. LaMOO is a multi-objective extension of recent works (Wang et al., 2020; Yang et al., 2021) that also learn space partitions but for a single black-box objective.

Empirically, LaMOO outperforms existing approaches on many benchmarks, including standard benchmarks in multi-objective black-box optimization, and real-world multi-objective problems like neural architecture search (NAS) (Cai et al., 2019; 2020) and molecule design. For example, as a meta-algorithm, LaMOO combined with CMA-ES as an inner routine requires only 62.5%, 8%, and 29% as many samples to reach the same hypervolume as the original CMA-ES (Igel et al., 2007a) in BraninCurrin (Belakaria et al., 2019), VehicleSafety (Liao et al., 2008) and Nasbench201 (Dong & Yang, 2020), respectively. On average, compared to qEHVI, LaMOO uses 50% samples to achieve the same performance in these problems. In addition, LaMOO with qEHVI (Daulton et al., 2020) and CMA-ES require 71% and 31% fewer samples on average, compared to naive qEHVI and CMA-ES, to achieve the same performance in molecule discovery.

## 2 RELATED WORK

**Bayesian Optimization (BO)** (Zitzler et al., 2003; Knowles, 2006; Ponweiser et al., 2008; Couckuyt et al., 2014; Paria et al., 2018; Yang et al., 2019; Daulton et al., 2020) is a popular family of methods to optimize black-box single and multi-objectives. Using observed samples, BO learns a surrogate model $\hat{f}(\mathbf{x})$, search for new promising candidates based on *acquisition function* built on $\hat{f}(\mathbf{x})$, and query the quality of these candidates with the ground truth black-box objective(s). In multi-objective Bayesian optimization (MOBO), most approaches leverage Expected Hypervolume Improvement

(EHVI) as their acquisition function (Zitzler et al., 2003; Couckuyt et al., 2014; Yang et al., 2019), since finding the Pareto frontier is equivalent to maximizing the hypervolume given a finite search space (Fleischer, 2003). There are methods (Knowles, 2006; Ponweiser et al., 2008; Paria et al., 2018) that use different acquisition functions like expected improvement (Jones et al., 1998) and Thompson sampling (Thompson, 1933). EVHI is computationally expensive: its cost increases exponentially with the number of objectives. To address this problem, qEHVI (Daulton et al., 2020) accelerates optimization by computing EHVI in parallel, and has become the state-of-the-art MOBO algorithm. In this paper, we leverage qEHVI as a candidate inner solver in our proposed LaMOO algorithm.

**Evolutionary algorithms (EAs)** (Deb et al., 2002a; Igel et al., 2007a; Zhang & Li, 2007; Beume et al., 2007; Fang et al., 2018) are also popular methods for MOO tasks. One category of MOO-EAs (Srinivas & Deb, 1994; Deb et al., 2002a; Deb & Jain, 2014) leverages Pareto dominance to simultaneously optimize all objectives. A second category (e.g., (Zhang & Li, 2007)) decomposes a multi-objective optimization problem into a number of single-objective sub-problems, converting a difficult MOO into several SOOs. Another category is quality indicator-based methods, such as (Beume et al., 2007) and (Igel et al., 2007a). They scalarize the current Pareto frontier using quality indicators (e.g., HV) and transfer a MOO to a SOO. New samples are generated by crossover and mutation operations from existing ones. However the drawbacks of non-quality indicator-based methods (i.e., the first two categories) can be not overlooked. Specifically, for MOO with many objectives, NSGA-II (Deb et al., 2002a) easily gets stuck in a dominance resistant solution (Pang et al., 2020) which is far from the true Pareto frontier. while MOEA/D perform better in MOO but how to specify the weight vector for problems with unknown Pareto front is the main challenge (Deb & Jain, 2014). In addition, A* search based algorithms are also considered to be extended to MOO (Stewart & White, 1991; Tung Tung & Lin Chew, 1992; De la Cruz et al., 2005).

**Quality Indicators**. Besides hypervolume, there are several other quality indicators (Van Veldhuizen & Lamont, 1998; Zitzler et al., 2000; Bosman & Thierens, 2003) for evaluating sample quality, which can be used to scalarize the MOO to SOO. The performance of a quality indicator can be evaluated by three metrics (Deng et al., 2007; Li et al., 2014), including convergence (closeness to the Pareto frontier), uniformity (the extent of the samples satisfy the uniform distribution), and spread (the extent of the obtained approximate Pareto frontier). Sec. B specifically illustrates the merits of each quality indicator. HyperVolume is the only metric we explored that can simultaneously satisfy the evaluation of convergence, uniformity, and spread without the knowledge of the true Pareto frontier while it may suffer from expensive calculation in many-objective problems. Therefore, throughout this work, we use HV to evaluate the optimization performance of different algorithms.

## 3 LEARNING SPACE PARTITIONS: A THEORETICAL UNDERSTANDING

Searching in high-dimensional space to find the optimal solution to a function is in general a challenging problem, especially when the function's properties are unknown to the search algorithm. The difficulty is mainly due to the curse of dimensionality: to adequately cover a $d$-dimensional space, in general, an exponential number of samples are needed.

For this, many works use a "coarse-to-fine" approach: partition the search space and then focusing on promising regions. Traditionally, manually defined criteria are used, e.g., axis-aligned partitions (Munos, 2011b), Voronoi diagrams (Kim et al., 2020), etc. Recently, (Wang et al., 2019; 2020; Yang et al., 2021) *learn* space partitions based on the data collected thus far, and show strong performance in NeurIPS black box optimization challenges (Sazanovich et al.; Kim et al.).

On the other hand, there is little quantitative understanding of space partition. In this paper, we first give a formal theoretical analysis on why learning plays an important role in space-partition approaches for SOO. Leveraging our understanding of how space partitioning works, we propose LaMOO which empirically outperforms existing SoTA methods on multiple MOO benchmarks.

### 3.1 PROBLEM SETTING

Intuitively, learning space partitions will yield strong performance if the classifier can determine which regions are promising given few data points. We formalize this intuition below and show why it is better than fixed and manually defined criteria for space partitioning.

Consider the following sequential decision task. We have $N$ samples in a *discrete* subset $S_0$ and there exists one sample $\mathbf{x}^*$ that achieves a minimal value of a scalar function $f$. Note that $f$ can be any property we want, e.g., in the Pareto optimal set. The goal is to construct a subset $S_T \subseteq S_0$ after $T$

steps, so that (1) $\mathbf{x}^* \in S_T$ and (2) $|S_T|$ is as small as possible. More formally, we define the reward function $r$ as the probability that we get $\mathbf{x}^*$ by randomly sampling from the resulting subset $S_T$:

$$r := \frac{1}{|S_T|} P(\mathbf{x}^* \in S_T) \tag{2}$$

It is clear that $0 \leq r \leq 1$. $r = 1$ means that we already found the optimal sample $\mathbf{x}^*$.

Here we use discrete case for simplicity and leave continuous case (i.e., partitioning a region $\Omega_0$ instead of a discrete set $S_0$) to future work. Note $N$ could be large, so here we consider it infeasible to enumerate $S_0$ to find $\mathbf{x}^*$. However, sampling from $S_0$, as well as comparing the quality of sampled solutions are allowed. An obvious baseline is to simply set $S_T := S_0$, then $r_{\mathrm{b}} = N^{-1}$. Now the question is: can we do better? Here we seek help from the following *oracle*:

**Definition 1** (($\alpha, \eta$)-Oracle). *Given a subset $S$ that contains $\mathbf{x}^*$, after taking $k$ samples from $S$, the oracle can find a* good *subset $S_{\mathrm{good}}$ with $|S_{\mathrm{good}}| \leq |S|/2$ and*

$$P\left(\mathbf{x}^* \in S_{\mathrm{good}} | \mathbf{x}^* \in S\right) \geq 1 - \exp\left(-\frac{k}{\eta|S|^\alpha}\right) \tag{3}$$

**Lemma 1.** *The algorithm to uniformly draw $k$ samples in $S$, pick the best and return is a $(1,1)$-oracle.*

See Appendix for proof. Note that a $(1,1)$-oracle is very weak, and is of little use in obtaining higher reward $r$. We typically hope for an oracle with smaller $\alpha$ and $\eta$ (i.e., both smaller than 1). Intuitively, such oracles are more sample-efficient: with few samples, they can narrow down the region containing the optimal solution $\mathbf{x}^*$ with high probability.

Note that $\alpha < 1$ corresponds to *semi-parametric models*. In these cases, the oracle has *generalization property*: with substantially fewer samples than $N$ (i.e., on the order of $N^\alpha$), the oracle is able to put the optimal solution $\mathbf{x}^*$ on the right side. In its extreme case when $\alpha = 0$ (or *parametric models*), whether we classify the optimal solution $\mathbf{x}^*$ on the correct side only depends on the *absolute* number of samples collected in $S$, and is independent of its size. For example, if the function to be optimized is linear, then with $d + 1$ samples, we can completely characterize the property of all $|S|$ samples.

**Relation with cutting plane.** Our setting can be regarded as a data-driven extension of cutting plane methods (Loganathan & Sherali, 1987; Vieira & Lisboa, 2019; Hinder, 2018) in optimization, in which a cutting plane is found at the current solution to reduce the search space. For example, if $f$ is convex and its gradient $\nabla f(\mathbf{x})$ is available, then we can set $S_{\mathrm{good}} := \{\mathbf{x} : \nabla f(\mathbf{x}_0)^\top (\mathbf{x} - \mathbf{x}_0) \leq 0, \mathbf{x} \in S_0\}$, since for any $\mathbf{x} \in S_0 \setminus S_{\mathrm{good}}$, convexity gives $f(\mathbf{x}) \geq f(\mathbf{x}_0) + \nabla f(\mathbf{x}_0)^\top (\mathbf{x} - \mathbf{x}_0) > f(\mathbf{x}_0)$ and thus $\mathbf{x}$ is not better than current $\mathbf{x}_0$. However, the cutting plane method relies on certain function properties like convexity. In contrast, learning space partition can leverage knowledge about the function forms, combined with observed samples so far, to better partition the space.

### 3.2 REWARDS UNDER OPTIMAL ACTION SEQUENCE

We now consider applying the $(\alpha, \eta)$-oracle iteratively for $T$ steps, by drawing $k_t$ samples from $S_{t-1}$ and setting $S_t := S_{\mathrm{good},t-1}$. We assume a total sample budget $K$, so $\sum_{t=1}^T k_t = K$. Note that $T \leq \log_2 N$ since we halve the set size with each iteration. Now the question is twofold. (1) How can we determine the action sequences $\{k_t\}$ in order to maximize the total reward $r$? (2) Following the optimal action sequences $\{k_t^*\}$, can $r^*$ be better than the baseline $r_{\mathrm{b}} = N^{-1}$? The answer is yes.

**Theorem 1.** *The algorithm yields a reward $r^*$ lower bounded by the following:*

$$r^* \geq r_{\mathrm{b}} \exp\left[\left(\log 2 - \frac{\eta N^\alpha \phi(\alpha, T)}{K}\right) T\right] \tag{4}$$

*where $r_{\mathrm{b}} := N^{-1}$ and $\phi(\alpha, T) := (1 - 2^{-\alpha T})/(1 - 2^{-\alpha})$.*

**Remarks.** Following Theorem 1, a key condition to make $r^* > r_{\mathrm{b}}$ is to ensure $\log 2 > \frac{\eta N^\alpha \phi(\alpha, T)}{K}$. This holds if when $\frac{\eta N^\alpha \phi(\alpha, T)}{K} \to 0$. Note that since $T \leq \log_2 N$, the final reward $r^*$ is upper bounded by 1 (rather than goes to $+\infty$). We consider some common practical scenarios below.

*Non-parametric models ($\alpha = 1$).* In this case, $\phi(\alpha, T) \leq 2$ and the condition becomes $\frac{1}{2}\log 2 > \eta N/K$. This happens when the total sample budget $K = \Theta(N)$, i.e., on the same order of $N$, which means that the partitioning algorithm obtains little advantage over exhaustive search.

*Semi-parametric models ($\alpha < 1$).* In this case, $\phi(\alpha, T) \leq 1/(1 - 2^{-\alpha})$ and the condition becomes $(1 - 2^{-\alpha}) \log 2 > \eta N^\alpha / K$. This happens when the total sample budget $K = \Theta(N^\alpha)$. In this case, we could use many fewer samples than exhaustive search to achieve better reward, thanks to the generalization property of the oracle.

*Parametric models ($\alpha = 0$).* Now $\phi(\alpha, T) = T$ and the condition becomes $\log 2 > \frac{\eta T}{K}$. Since $T \leq \log_2 N$, the total sample budget can be set to be $K = \Theta(\log N)$. Intuitively, the algorithm performs iterative halving (or binary search) to narrow down the search toward promising regions.

### 3.3 EXTENSION TO MULTI-OBJECTIVE OPTIMIZATION

Given our understanding of space partitioning, we now extend this idea to MOO. Intuitively, we want "good" regions to be always picked by the space partition. For SOO, it is possible since the optimal solution is a single point. How about MOO?

Unlike SOO, in MOO we aim for a continuous region, the *Pareto optimal set* $\Omega_P := \{\mathbf{x} : \nexists \mathbf{x}' \neq \mathbf{x} : \mathbf{f}(\mathbf{x}') \prec \mathbf{f}(\mathbf{x})\}$. A key variable is the regularity of $\Omega_P$: if it is highly non-regular and not captured by a simple partition boundary (ideally a parametric boundary), then learning a space partition would be difficult. Interestingly, the shape of $\Omega_P$ can be characterized for quadratic objectives:

**Observation 1.** *If all $f_j$ are isotropic, $f_j(\mathbf{x}) = \|\mathbf{x} - \mathbf{c}_j\|_2^2$, then $\Omega_P = \text{ConvexHull}(\mathbf{c}_1, \ldots, \mathbf{c}_q)$.*

**Observation 2.** *If $M = 2$ and $f_j(\mathbf{x}) = (\mathbf{x} - \mathbf{c}_j)^\top H_j (\mathbf{x} - \mathbf{c}_j)$ where $H_j$ are positive definite symmetric matrices, then there exists $\mathbf{w}_1 := H_2(\mathbf{c}_2 - \mathbf{c}_1)$ and $\mathbf{w}_2 := H_1(\mathbf{c}_1 - \mathbf{c}_2)$, so that for any $\mathbf{x} \in \Omega_P$, $\mathbf{w}_1^\top (\mathbf{x} - \mathbf{c}_1) \geq 0$ and $\mathbf{w}_2^\top (\mathbf{x} - \mathbf{c}_2) \geq 0$.*

In both cases, $\Omega_P$ can be separated from non-Pareto regions $\Omega \backslash \Omega_P$ via a linear hyperplane. Empirically, $\Omega_P$ only occupies a small region of the entire search space (Sec. 4), and quickly focusing samples on the promising regions is critical for high sample efficiency.

In the general case, characterizing $\Omega_P$ is analytically hard and requires domain knowledge about the objectives (Li et al., 2014). However, for MOO algorithms in practice, knowing that $\Omega_P$ can be separated from $\Omega \backslash \Omega_P$ via simple decision planes is already useful: we could learn such decision planes given previous data that are already collected, and sample further in promising regions.

## 4 LAMOO: LATENT ACTION MULTI-OBJECTIVE OPTIMIZATION

In Sec. 3, for convenience, we only analyze a greedy approach, which makes decisions on space partitions and never revises them afterwards. While this greedy approach indeed works (as shown in Sec. 5.3), an early incorrect partition could easily rule out regions that turn out to be good but weren't identified with few samples. In practice, we want to keep the decision softer: while *exploiting* the promising region, we also *explore* regions that are currently believed to be sub-optimal given limited samples. It is possible that these regions turn out to contain good solutions when more samples are available, and the oracle can then make a different partition.

To balance the trade-off between exploration and exploitation to cope with the generalization error of the learned classifier, we leverage Monte Carlo Tree Search (MCTS) (Kocsis & Szepesvári, 2006) and propose our algorithm LaMOO. As shown in Alg. 1, LaMOO has four steps: (1) learn to partition the search space given previous observed data points $D_t$, which are collected $\{\mathbf{x}_i, \mathbf{f}(\mathbf{x}_i)\}$ from iterations 0 to $t$. (2) With this information, we partition the region into promising and non-promising regions, and learn a classifier $h(\cdot)$ to separate them. (3) We select the region to sample from, based on the UCB value of each node. (4) We sample selected regions to obtain future data points $D_{t+1}$.

**Learning Space Partitions**. We construct the partition oracle using the *dominance number*. Let $D_t$ be the collected samples up to iteration $t$ and $D_{t,j} := D_t \cap \Omega_j$ be the samples within the region $\Omega_j$ we want to partition. For each sample $\mathbf{x} \in D_{t,j}$, its dominance number $o_{t,j}(\mathbf{x})$ at iteration $t$ is defined as the number of samples in $\Omega_j$ that dominate $\mathbf{x}$ (here $\mathbb{I}[\cdot]$ is the indicator function):

$$o_{t,j}(\mathbf{x}) := \sum_{\mathbf{x}_i \in D_{t,j}} \mathbb{I}[\mathbf{x} \prec_{\mathbf{f}} \mathbf{x}_i, \ \mathbf{x} \neq \mathbf{x}_i] \tag{5}$$

While naive computation requires $O(|D_{t,j}|^2)$ operations, we use Maxima Set (Kung et al., 1975) which runs in $O(|D_{t,j}| \log |D_{t,j}|)$. For $\mathbf{x} \in \Omega_P$, $o(\mathbf{x}) = 0$.

---

**Algorithm 1** LaMOO Pseudocode.

1: **Inputs:** Initial $D_0$ from uniform sampling, sample budget $T$.
2: **for** $t = 0, \ldots, T$ **do**
3:     Set $\mathcal{L} \leftarrow \{\Omega_{\text{root}}\}$ (collections of regions to be split).
4:     **while** $\mathcal{L} \neq \emptyset$ **do**
5:         $\Omega_j \leftarrow \text{pop\_first\_element}(\mathcal{L})$, $D_{t,j} \leftarrow D_t \cap \Omega_j$, $n_{t,j} \leftarrow |D_{t,j}|$.
6:         Compute dominance number $o_{t,j}$ of $D_{t,j}$ using Eqn. 5 and train SVM model $h(\cdot)$.
7:         **If** $(D_{t,j}, o_{t,j})$ is splittable by SVM, **then** $\mathcal{L} \leftarrow \mathcal{L} \cup \text{Partition}(\Omega_j, h(\cdot))$.
8:     **end while**
9:     **for** $k = \text{root}$, $k$ is not leaf node **do**
10:         $D_{t,k} \leftarrow D_t \cap \Omega_k$, $v_{t,k} \leftarrow \text{HyperVolume}(D_{t,k})$, $n_{t,k} \leftarrow |D_{t,k}|$.
11:         $k \leftarrow \arg \max\limits_{c \,\in\, \text{children}(k)} \text{UCB}_{t,c}$, where $\text{UCB}_{t,c} := v_{t,c} + 2C_p\sqrt{\frac{2\log(n_{t,k})}{n_{t,c}}}$
12:     **end for**
13:     $D_{t+1} \leftarrow D_t \cup D_{\text{new}}$, where $D_{\text{new}}$ is drawn from $\Omega_k$ based on qEHVI or CMA-ES.
14: **end for**

---

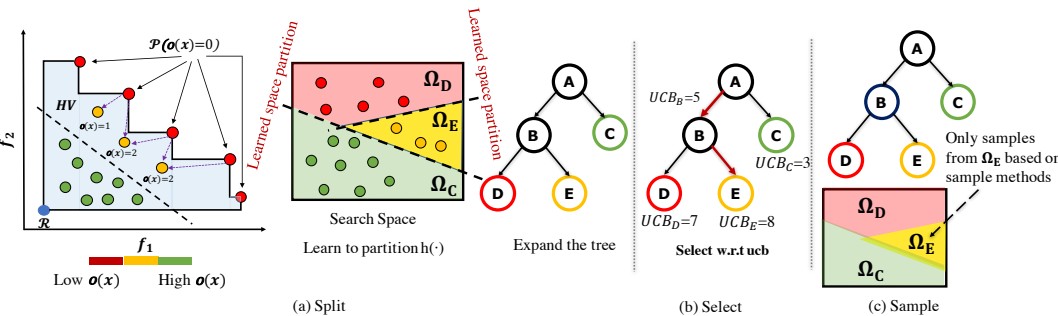

Figure 2: (a) The leaf nodes D and E that correspond to the non-splittable space $\Omega_D$ and $\Omega_E$. (b). The node selection procedure based on the UCB value. (c). The new samples generation from the selected space $\Omega_E$ for bayesian optimization.

For each $D_{t,j}$, we then get good (small $o(\mathbf{x})$) and bad (large $o(\mathbf{x})$) samples by ranking them according to $o(\mathbf{x})$. The smallest 50% are labeled to be positive while others are negative. Based on the labeled samples, a classifier (e.g., Support Vector Machine (SVM)) is trained to learn a decision boundary as the latent action. We choose SVM since the classifier needs to be decent in regions with few samples, and has the flexibility of being parametric or non-parametric.

**Exploration Using Upper Confidence Bounds(UCB)**. As shown in Fig. 2, LaMOO selects the final leaf node by always choosing the child node with larger UCB value. The *UCB value* for a node $j$ is defined as $\text{UCB}_j := v_j + 2C_p\sqrt{2\log n_{\text{parent(j)}}/n_j}$, where $n_j$ is the number of samples in node $j$, $C_p$ is a tunable hyperparameter which controls the degree of exploration, and $v_j$ is the hypervolume of the samples in node $j$. The selected leaf corresponds the partitioned region $\Omega_k$ as shown in Alg. 1.

**Sampling in Search Region**. We use existing algorithms as a sampling strategy in a leaf node, e.g., qEHVI (Daulton et al., 2020)) and CMA-ES (Igel et al., 2007a). Therefore, LaMOO can be regarded as a *meta-algorithm*, applicable to any existing SOO/MOO solver to boost its performance.

LaMOO with qEHVI. As a multi-objective solver, qEHVI finds data points to maximize a parallel version of Expected Hypervolume Improvement (EHVI) via Bayesian Optimization (BO). To incorporate qEHVI into LaMOO's sampling step, we confine qEHVI's search space using the tree-structured partition to better search MOO solutions.

LaMOO with CMA-ES. CMA-ES is an evolutionary algorithm (EA) originally designed for single-objective optimization. As a leaf sampler, CMA-ES is used to pick a sample that maximizes the dominance number $o(\mathbf{x})$ within the leaf. Since $o(\mathbf{x})$ changes over iterations, at iteration $t$, we first update $o_{t'}(\mathbf{x})$ of all previous samples at $t' < t$ to $o_t(\mathbf{x})$, then use CMA-ES. Similar to the qEHVI case, we constrain our search to be within the leaf region.

Once a set of new samples $D_{\text{new}}$ is obtained (as well as its multiple function values $\mathbf{f}(D_{\text{new}})$), we update all partitions along its path and the entire procedure is repeated.

## 5 EXPERIMENTS

We evaluate the performance of LaMOO in a diverse set of scenarios. This includes synthetic functions, and several real-world MOO problems like neural architecture search, automobile safety design, and molecule discovery. In such real problems, often a bunch of criteria needs to be optimized at the same time. For example, for molecule (drug) discovery, one wants the designed drug to be effective towards the target disease, able to be easily synthesized, and be non-toxic to human body.

### 5.1 SMALL-SCALE PROBLEMS

**Synthetic Functions.** Branin-Currin (Belakaria et al., 2019) is a function with 2-dimensional input and 2 objectives. DTLZ2 (Deb et al., 2002b) is a classical scalable multi-objective problem and is popularly used as a benchmark in the MOO community. We evaluate LaMOO as well as baselines in DTLZ2 with 18 dimensions and 2 objectives, and 12 dimensions and 10 objectives, respectively.

**Structural Optimization in Automobile Safety Design (vehicle safety)** is a real-world problem with 5-dimensional input and 3 objectives, including (1) the mass of the vehicle, (2) the collision acceleration in a full-frontal crash, and (3) the toe-board intrusion (Liao et al., 2008).

**Nasbench201** is a public benchmark to evaluate NAS algorithms (Dong & Yang, 2020). There are 15625 architectures in Nasbench201, with groundtruth #FLOPs and accuracy in CI-FAR10 (Krizhevsky, 2009). Our goal is to minimize #FLOPs and maximize accuracy in this search space. We normalized #FLOPs to range $[-1, 0]$ and accuracy to $[0, 1]$.

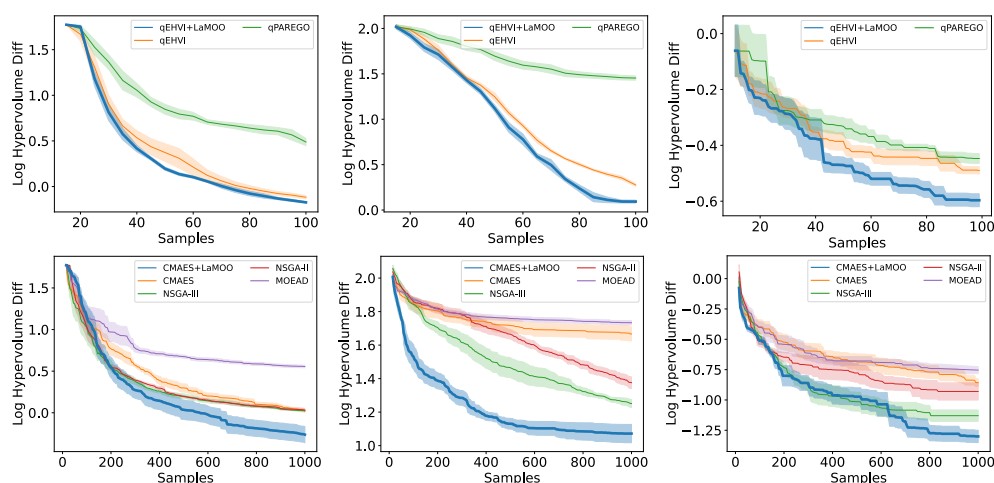

Figure 3: **Left**: Branin-Currin with 2 dimensions and 2 objectives. **Middle**: VehicleSafety with 5 dimensions and 3 objectives. **Right**: Nasbench201 with 6 dimensions and 2 objectives. We ran each algorithm 7 times (shaded area is $\pm$ std of the mean). **Top**: Bayesian Optimization w/o LaMOO. **Bottom**: evolutionary algorithms w/o LaMOO. Note the two algorithm families show very different sample efficiency in MOO tasks.

We compare LaMOO with 4 classical evolutionary algorithms (CMA-ES (Igel et al., 2007a), MOEA/D (Zhang & Li, 2007), NSGA-II (Deb et al., 2002a), and NSGA-III (Deb & Jain, 2014)) and 2 state-of-the-art BO methods (qEHVI (Daulton et al., 2020) and qParego (Knowles, 2006)).

**Evaluation Criterion**. we first obtain the maximal hypervolume (either by ground truth or from the estimation of massive sampling), then run each algorithm and compute the log hypervolume difference (Daulton et al., 2020):

$$HV_{\text{log\_diff}} := \log(HV_{\text{max}} - HV_{\text{cur}}) \qquad (6)$$

where $HV_{\text{cur}}$ is the hypervolume of current samples obtained by the algorithm with given budget.

**Result.** As shown in Fig. 3, LaMOO with qEHVI outperforms all our BO baselines and LaMOO with CMA-ES outperforms all our EA baselines, in terms of $HV_{\text{log\_diff}}$.

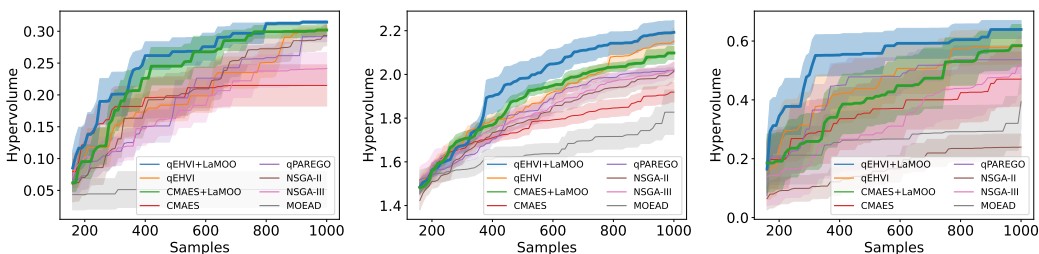

Figure 4: DTLZ2 with many objectives, We ran each algorithm 7 times (shaded area is $\pm$ std of the mean). From **left** to **right**: BO with 2 objectives; EA with 2 objectives; BO with 10 objectives; EA with 10 objectives.

Evolutionary algorithms rely on mutation and crossover of previous samples to generate new ones, and may be trapped into local optima. Thanks to MCTS, LaMOO also considers exploration and greatly improves upon vanilla CMA-ES over three different tasks with 1000/200 samples in small-scale /many objective problems. In addition, by plugging in BO, LaMOO+qEHVI achieves 225% sample efficiency compared to other BO algorithms on Nasbench201. This result indicates that for high-dimensional problems (6 in Nasbench201), space partitioning leads to faster optimization. We further analyze very high-dimensional problems on Sec. 5.2 and visualize Pareto frontier in Fig. 12.

**Optimization of Many Objectives**. While NSGA-II and NSGA-III perform well in the two-objective problems, all evolutionary-based baselines get stuck in the ten-objective problems. In contrast, LaMOO performs reasonably well. From Fig. 4, qEHVI+LaMOO shows strong performance in ten objectives. When combined with a CMA-ES, LaMOO helps it escape the initial region to focus on a smaller promising region by space partitioning.

## 5.2 MULTI-OBJECTIVE MOLECULE DISCOVERY

Figure 5: Molecule Discovery: **Left**: Molecule discovery with two objectives (GSK3$\beta$+JNK3). **Middle**: Molecule discovery with three objectives (QED+SA+SARS). **Right**: Molecule Discovery with four objectives (GSK3$\beta$+JNK3+QED+SA). We ran each algorithm 15 times (shaded area is $\pm$ std of the mean).

Next, we tackle the practical problem of multi-objective molecular generation, which is a high-dimensional problem (search space is 32-dimensional). Molecular generation models are a critical component of pharmaceutical drug discovery, wherein a cheap-to-run *in silico* model proposes promising molecules which can then be synthesized and tested in a lab (Vamathevan et al., 2019). However, one commonly requires the generated molecule to satisfy multiple constraints: for example, new drugs should generally be non-toxic and ideally easy-to-synthesize, in addition to their primary purpose. Therefore, in this work, we consider several multi-objective molecular generation setups from prior work on molecular generation (Yu et al., 2019; Jin et al., 2020b; Yang et al., 2021): (1) activity against biological targets GSK3$\beta$ and JNK3, (2) the same targets together with QED (a standard measure of "drug-likeness") and SA (a standard measure of synthetic accessibility), and (3) activity against SARS together with QED and SA. In each task, we propose samples from a pre-trained 32-dimensional latent space from (Jin et al., 2020a), which are then decoded into molecular strings and fed into the property evaluators from prior work.

Fig. 5 shows that LaMOO+qEHVI outperforms all baselines by up to 10% on various combinations of objectives. While EA struggles to optimize these high-dimensional problems due to the limitations mentioned in Sec. 2, LaMOO helps them (e.g., CMA-ES) to perform much better.

## 5.3 ABLATION STUDIES

**Visualization of LaMOO.** To understand how LaMOO works, we visualize its optimization procedure for Branin-Currin. First, the Pareto optimal set $\Omega_P$ is estimated from $10^6$ random samples

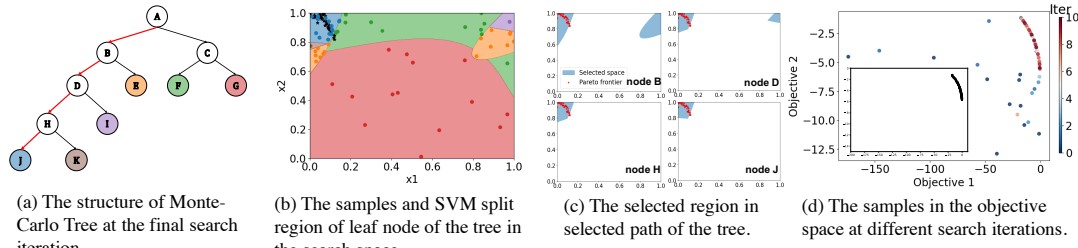

(a) The structure of Monte-Carlo Tree at the final search iteration.

(b) The samples and SVM split region of leaf node of the tree in the search space.

(c) The selected region in selected path of the tree.

(d) The samples in the objective space at different search iterations.

Figure 6: Visualization of selected region at different search iterations and nodes. (a) The Monte-Carlo tree with colored leaves. Selected path is marked in red. (b) Visualization of the regions($\Omega_J, \Omega_K, \Omega_I, \Omega_E, \Omega_F, \Omega_G$) that are consistent with leaves in (a) in the search space. (c) Visualization of selected path at final iteration. (d) Visualization of samples during search; bottom left is the Pareto frontier estimated from one million samples.

(marked as black stars), as shown in both search and objective space (Fig. 6(b) and bottom left of Fig. 6(c)). Over several iterations, LaMOO progressively prunes away unpromising regions so that the remaining regions approach $\Omega_P$ (Fig 6(c)). Fig 6(a) shows the final tree structure. The color of each leaf node corresponds to a region in the search space (Fig 6(b)). The selected region is recursively bounded by SVM classifiers corresponding to nodes on the selected path (red arrow in Fig 6(a)). The new samples are only generated from the most promising region $\Omega_J$, improving sample efficiency.

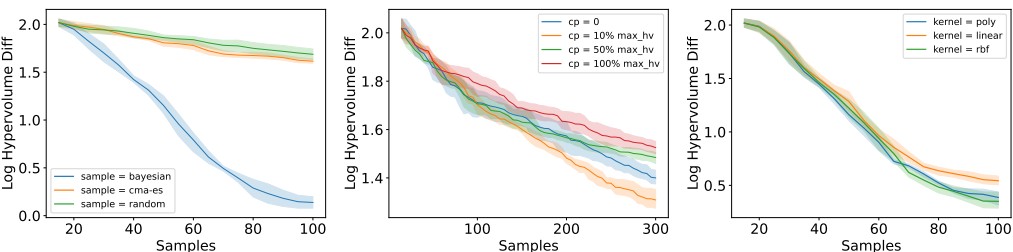

Figure 7: Ablation studies on hyperparameters and sampling methods in LaMOO. **Left**: Sampling without Bayesian/CMA-ES. **Middle**: Sampling with different $C_p$. **Right**: Partitioning with different svm kernels

**Ablation of Design Choices.** We show how different hyperparameters and sampling methods play a role in the performance. We perform the study in VehicleSafety below.

*Sampling methods.* LaMOO can be integrated with different sample methods, including Bayesian Optimization (e.g., qEHVI) and evolutionary algorithms (e.g., CMA-ES). Fig. 7(left) shows that compared to random sampling, qEHVI improves a lot while CMA-ES only improves slightly. This is consistent with our previous finding that for MOO, BO is much more efficient than EA.

*The exploration factor* $C_p$ controls the balance of exploration and exploitation. A larger $C_p$ guides LaMOO to visit the sub-optimal regions more often. Based on the results in Fig. 7(middle), greedy search ($C_p = 0$) leads to worse performance compared to a proper $C_p$ value (i.e. 10% of maximum hypervolume), which justifies our usage of MCTS. On the other hand, over-exploration can also yield even worse results than greedy search. Therefore, a "rule of thumb" is to set the $C_p$ to be roughly 10% of the maximum hypervolume $HV_{\max}$. When $HV_{\max}$ is unknown, $C_p$ can be set empirically.

*SVM kernels.* As shown in Fig. 7(right), we find that the RBF kernel performs the best, in agreement with (Wang et al., 2020). Thanks to the non-linearity of the polynomial and RBF kernels, their region partitions perform better compared to a linear one.

## 6 CONCLUSION

We propose a search space partition optimizer called LaMOO as a meta-algorithm that extends prior single-objective works (Wang et al., 2020; Yang et al., 2021) to multi-objective optimization. We demonstrated both theoretically and via experiments on multiple MOO tasks that LaMOO significantly improves the search performance compared to strong baselines like qEHVI and CMA-ES.

## 7  ACKNOWLEDGEMENT

This work was supported in part by NSF Grants #1815619 and #2105564, a VMWare grant, and computational resources supported by the Academic & Research Computing group at Worcester Polytechnic Institute.

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

## A PROOFS

**Lemma 1.** *The algorithm to uniformly draw $k$ samples in $S$, pick the best and return is a $(1, 1)$-oracle.*

*Proof.* Consider the following simple $(1, 1)$-oracle for single-objective optimization: after sampling $k$ samples, we rank them according to their function values, and split them into two $k/2$ smaller subsets $\tilde{S}_{\text{good}}$ and $\tilde{S}_{\text{bad}}$. Other points are randomly assigned to either of the two subsets. Then if $\mathbf{x}^*$ happens to be among the $k$ collected samples (which happens with probability $k/|S|$), definitely we have $\mathbf{x}^* \in S_{\text{good}}$. Therefore, we have:

$$P\left(\mathbf{x}^* \in S_{\text{good}} | \mathbf{x}^* \in S\right) \geq \frac{k}{|S|} \geq 1 - \exp\left(-\frac{k}{|S|}\right) \tag{7}$$

which is an oracle with $\alpha = \eta = 1$. The last inequality is due to $e^x \geq 1 + x$ (and thus $e^{-x} \geq 1 - x$). $\square$

**Lemma 2.** *Define $g(\lambda) : \mathbb{R}^+ \mapsto \mathbb{R}^+$ as:*

$$g(\lambda) : \lambda \mapsto \sum_{t=1}^{T} w_t \log\left(1 + \frac{1}{\lambda w_t}\right) \tag{8}$$

*The following maximization problem*

$$\max_{\{z_t\}} \sum_{t=1}^{T} \log\left(1 - e^{-z_t}\right) \quad \text{s.t.} \quad \sum_{t=1}^{T} w_t z_t = K \tag{9}$$

*has optimal solutions*

$$z_t^* = \log\left(1 + \frac{1}{\lambda w_t}\right), \quad 1 \leq t \leq T \tag{10}$$

*where $\lambda$ is determined by $g(\lambda) = K$. With optimal $\{z_t^*\}$, the objective reaches $-\sum_t \log(1 + \lambda w_t)$.*

*Proof.* Its Lagrange is:

$$J\left(\{z_t\}\right) = \sum_{t=1}^{T} \log\left(1 - e^{-z_t}\right) - \lambda\left(\sum_{t=1}^{T} w_t z_t - K\right) \tag{11}$$

Taking derivative w.r.t. $z_t$ and we have:

$$
\begin{aligned}
\frac{\partial J}{\partial z_t} &= \frac{e^{-z_t}}{1 - e^{-z_t}} - \lambda w_t = 0. \\
\frac{1}{1 - e^{-z_t}} - 1 - \lambda w_t &= 0 \\
\frac{1}{1 - e^{-z_t}} &= 1 + \lambda w_t \\
1 - e^{-z_t} &= \frac{1}{1 + \lambda w_t} \\
e^{-z_t} = 1 - \frac{1}{1 + \lambda w_t} &= \frac{\lambda w_t}{1 + \lambda w_t} \\
z_t = -\log\frac{\lambda w_t}{1 + \lambda w_t} = \log\frac{1 + \lambda w_t}{\lambda w_t} &= \log\left(1 + \frac{1}{\lambda w_t}\right)
\end{aligned}
\tag{12}
$$

$\square$

**Lemma 3.** *Both $g(\lambda)$ and $g^{-1}(y)$ are monotonously decreasing. Furthermore, let $\bar{w} := T\left(\sum_{t=1}^{T} w_t^{-1}\right)^{-1}$ be the Harmonic mean of $\{w_t\}$ and $w_{\max} := \max_{t=1}^{T} w_t$, we have:*

$$\frac{\bar{w}^{-1}}{\exp(\bar{w}^{-1}y/T) - 1} \leq g^{-1}(y) \leq \frac{w_{\max}^{-1}}{\exp(w_{\max}^{-1}y/T) - 1} \leq \frac{T}{y}. \tag{13}$$

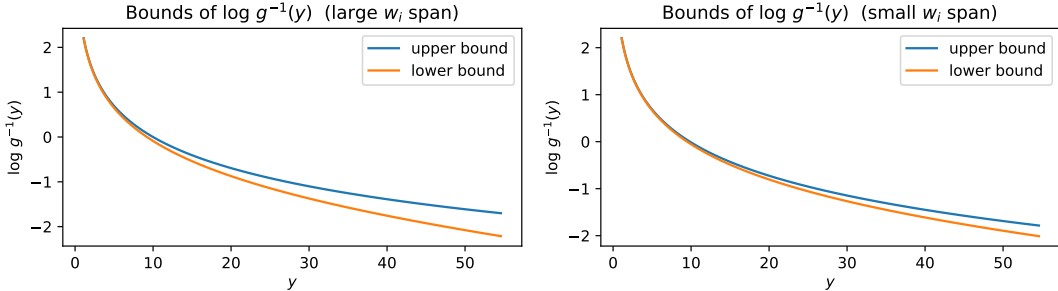

Figure 8: Upper and lower bounds of $g^{-1}(y)$ with different $\{w_i\}$. Left: $w_i = 2^{\text{linspace}(-0.1, 10)}$. Right: $w_i = 2^{\text{linspace}(2,5)}$. Small $\{w_i\}$ span leads to better bounds.

*Proof.* It is easy to see when $\lambda$ increases, each term in $g(\lambda)$ decreases and thus $g(\lambda)$ is a decreasing function of $\lambda$. Therefore, its inverse mapping $g^{-1}(y)$ is also decreasing.

Let $\mu(y) := 1/g^{-1}(y) > 0$. Then we have:

$$y = \sum_{t=1}^{T} w_t \log\left(1 + \frac{\mu(y)}{w_t}\right) \tag{14}$$

It is clear that when $y = 0$, $\mu(y) = 0$. Taking derivative with respect to $y$ in both side, we have:

$$1 = \mu'(y) \sum_{t=1}^{T} \frac{1}{1 + \frac{\mu(y)}{w_t}} \tag{15}$$

where $\mu'(y) = \frac{d\mu(y)}{dy}$ is the derivative of $\mu(y)$. Using the property of Harmonic mean, we have:

$$\mu'(y) = \left(\sum_{t=1}^{T} \frac{1}{1 + \frac{\mu(y)}{w_t}}\right)^{-1} \leq \frac{\sum_{t=1}^{T} 1 + \frac{\mu(y)}{w_t}}{T^2} = \frac{1}{T}\left(1 + \frac{\mu(y)}{\bar{w}}\right) \tag{16}$$

This gives:

$$\frac{\mu'(y)}{1 + \mu(y)/\bar{w}} \leq \frac{1}{T} \tag{17}$$

Integrate on both side starting from $y = 0$, we have:

$$\left.\bar{w} \log(1 + \mu(y)/\bar{w})\right|_0^y \leq \left.\frac{y}{T}\right|_0^y \tag{18}$$

Using $\mu(0) = 0$ we thus have:

$$\bar{w} \log(1 + \mu(y)/\bar{w}) \leq \frac{y}{T} \tag{19}$$

This leads to $\mu(y) \leq \bar{w}\left[\exp(y\bar{w}^{-1}/T) - 1\right]$. With $g^{-1}(y) = 1/\mu(y)$, we arrive the final lower bound for $g^{-1}(y)$.

For an alternative upper bound of $g^{-1}(y)$, we just notice that (here $w_{\max} := \max_t w_t$):

$$\mu'(y) = \left(\sum_{t=1}^{T} \frac{1}{1 + \frac{\mu(y)}{w_t}}\right)^{-1} \geq \left(\frac{T}{1 + \frac{\mu(y)}{w_{\max}}}\right)^{-1} = \frac{1}{T}\left(1 + \frac{\mu(y)}{w_{\max}}\right) \tag{20}$$

Using the same technique as above, we have $\mu(y) \geq w_{\max}\left[\exp(yw_{\max}^{-1}/T) - 1\right]$ and the upper bound of $g^{-1}(y)$ follows.

Finally, note that $e^x \geq 1 + x$, we have

$$\frac{w_{\max}^{-1}}{\exp(w_{\max}^{-1}y/T) - 1} \leq \frac{w_{\max}^{-1}}{w_{\max}^{-1}y/T} = \frac{T}{y} \tag{21}$$

$\square$

**Theorem 1.** *Following optimal sequence, the algorithm yields a reward $r^*$ lower bounded by the following:*

$$r^* \geq r_{\mathrm{b}} \exp\left[\left(\log 2 - \frac{\eta N^\alpha \phi(\alpha, T)}{K}\right) T\right] \tag{22}$$

*where $r_{\mathrm{b}} := N^{-1}$ and $\phi(\alpha, T) := (1 - 2^{-\alpha T})/(1 - 2^{-\alpha})$.*

*Proof.* First note that $|S_T| \leq |S_0|/2^T$ and thus $\frac{1}{|S_T|} \geq 2^T/N$. So we just need to bound $P(\mathbf{x}^* \in S_T)$, which can be written as:

$$P(\mathbf{x}^* \in S_T) = \prod_{t=1}^{T} P(\mathbf{x}^* \in S_t | \mathbf{x}^* \in S_{t-1}) \geq \prod_{t=1}^{T} \left(1 - \exp\left(-\frac{k_t}{\eta|S_{t-1}|^\alpha}\right)\right) \tag{23}$$

Therefore we have

$$\log P(\mathbf{x}^* \in S_T) \geq \sum_{t=1}^{T} \log\left(1 - \exp\left(-\frac{k_t}{\eta|S_{t-1}|^\alpha}\right)\right) \tag{24}$$

We want to find the action sequence $\{k_t\}$ so that $\log P(\mathbf{x}^* \in S_T)$ is maximized. Let $w_t := \eta|S_{t-1}|^\alpha$ and $z_t := k_t/w_t$, applying Lemma 2, and we know that

$$\max_{\{k_t\}} \log P(\mathbf{x}^* \in S_T) \geq -\sum_{t=1}^{T} \log(1 + \lambda w_t) \tag{25}$$

where the Lagrangian multiplier $\lambda$ satisfies the equation $g(\lambda) = K$.

Now we have:

$$\sum_{t=1}^{T} \log(1 + \lambda w_t) \overset{①}{\leq} \sum_{t=1}^{T} \log\left(1 + \frac{T}{K} w_t\right) \tag{26}$$

$$\overset{②}{\leq} \sum_{t=1}^{T} \log\left(1 + \frac{T}{K}\eta(N/2^{t-1})^\alpha\right) \tag{27}$$

$$\overset{③}{\leq} \frac{\eta T N^\alpha}{K} \sum_{t=1}^{T} \frac{1}{2^{\alpha(t-1)}} \tag{28}$$

$$= \phi(\alpha, T)\frac{\eta T N^\alpha}{K} \tag{29}$$

Here ① is due to Lemma 3 which tells that $\lambda = g^{-1}(K) \leq T/K$, ② is due to $w_t := \eta|S_{t-1}|^\alpha$ and $|S_{t-1}| \leq N/2^{t-1}$, and ③ due to $\log(1 + x) \leq x$.

Putting all of them together, we know that

$$r^* \geq \max_{\{k_t\}} \frac{1}{|S_T|} P(\mathbf{x}^* \in S_T) \geq \frac{2^T}{N} \exp\left(-\phi(\alpha, T)\frac{\eta T N^\alpha}{K}\right) \tag{30}$$

$\square$

**Optimal action sequence** $\{k_t^*\}$. From the proof, we could also write down the optimal action sequence that achieves the best reward: $k_t^* = w_t \log\left(1 + \frac{1}{\lambda w_t}\right)$, where $w_t := \eta|S_{t-1}|^\alpha$. Using Lemma 3, we could compute the upper and lower bound estimation of $\lambda = g^{-1}(K)$. Here $\bar{w} := T\left(\sum_{t=1}^{T} w_t^{-1}\right)^{-1}$ be the Harmonic mean of $\{w_t\}$ and $w_{\max} := \max_{t=1}^{T} w_t$:

$$\frac{\bar{w}^{-1}}{\exp(\bar{w}^{-1}K/T) - 1} \leq \lambda \leq \frac{w_{\max}^{-1}}{\exp(w_{\max}^{-1}K/T) - 1} \tag{31}$$

With $\lambda$, we could compute approximate $\{k_t^*\}$. Here we make a rough estimation of $\{k_t^*\}$ if we terminate the algorithm when $|S_T|$ is still fairly large. This case corresponds to the setting $T =$

$\beta \log_2 N$ where $\beta < 1$ and all $w_t \sim N^\alpha$. With $K = \Theta(N^\alpha)$ as in semi-parametric case, $\bar{w}^{-1} K = \Theta(1)$, $\exp(\bar{w}^{-1} K / T) - 1 \approx \bar{w}^{-1} K / T$ and $\lambda w_t \sim \log_2 N \gg 1$. Since $\log(1 + x) \approx x$ for small $x$, we have $k_t^* \approx w_t \frac{1}{\lambda w_t} = 1/\lambda$, which is independent of $t$. Therefore, a constant amount of sampling at each stage is approximately optimal.

**Observation 1.** *If all $f_j$ are isotropic, $f_j(\mathbf{x}) = \|\mathbf{x} - \mathbf{c}_j\|_2^2$, then $\Omega_P = \text{ConvexHull}(\mathbf{c}_1, \ldots, \mathbf{c}_M)$.*

*Proof.* Consider $J(\mathbf{x}; \mu) := \sum_{j=1}^M \mu_j f_j(\mathbf{x})$ where the weights $\mu_j \geq 0$ satisfies $\sum_j \mu_j = 1$. For brevity, we write the constraint as $\Delta := \{\mu : \mu_j \geq 0, \sum_j \mu_j = 1\}$.

Now consider the Pareto Set $\Omega_P := \{\mathbf{x} : \exists \mu \in \Delta : \nabla_\mathbf{x} J(\mathbf{x}; \mu) = 0\}$. We have the following:

$$\nabla_\mathbf{x} J(\mathbf{x}; \mu) = 0 \tag{32}$$

$$\iff \sum_j \mu_j \nabla_\mathbf{x} f_j(\mathbf{x}) = 0 \tag{33}$$

$$\iff \sum_j \mu_j (\mathbf{x} - \mathbf{c}_j) = 0 \tag{34}$$

$$\iff \mathbf{x} = \frac{\sum_j \mu_j \mathbf{c}_j}{\sum_j \mu_j} = \sum_j \mu_j \mathbf{c}_j \tag{35}$$

The last step is due to the fact that $\sum_j \mu_j = 1$. Therefore, for any $\mathbf{x} \in \Omega_P$, $\mathbf{x}$ is a convex combination of $\{\mathbf{c}_1, \ldots, \mathbf{c}_M\}$ and thus $\mathbf{x} \in \text{ConvexHull}(\mathbf{c}_1, \ldots, \mathbf{c}_M)$. Conversely, for any $\mathbf{x} \in \text{ConvexHull}(\mathbf{c}_1, \ldots, \mathbf{c}_M)$, we know $\nabla_\mathbf{x} J(\mathbf{x}; \mu) = 0$ and thus $\mathbf{x} \in \Omega_P$. $\square$

**Observation 2.** *If $M = 2$ and $f_j(\mathbf{x}) = (\mathbf{x} - \mathbf{c}_j)^\top H_j (\mathbf{x} - \mathbf{c}_j)$ where $H_j$ are positive definite symmetric matrices, then there exists $\mathbf{w}_1 := H_2(\mathbf{c}_2 - \mathbf{c}_1)$ and $\mathbf{w}_2 := H_1(\mathbf{c}_1 - \mathbf{c}_2)$, so that for any $\mathbf{x} \in \Omega_P$, $\mathbf{w}_1^\top (\mathbf{x} - \mathbf{c}_1) \geq 0$ and $\mathbf{w}_2^\top (\mathbf{x} - \mathbf{c}_2) \geq 0$.*

*Proof.* Following Observation 1, similarly we have for all $x \in \Omega_P$, $\sum_j \mu_j H_j (\mathbf{x} - \mathbf{c}_j) = 0$, which gives:

$$\mathbf{x} = \left( \sum_j \mu_j H_j \right)^{-1} \sum_j \mu_j H_j \mathbf{c}_j \tag{36}$$

Note that this is an expression of the Pareto Set $\Omega_P$.

Let $A_j := (\sum_j \mu_j H_j)^{-1} \mu_j H_j$. Then $\sum_j A_j = I$. Note that while $\sum_j \mu_j H_j$ and $(\sum_j \mu_j H_j)^{-1}$ are positive definite matrix. $A_j$ may not be.

Let $M := \sum_i \mu_i H_i$. Since $\mu \in \Delta$, $M$ is a PD matrix. Note that we have

$$\sum_j \mu_j H_j \mathbf{c}_j = \sum_j \mu_j H_j \mathbf{c}_j - \sum_j \mu_j H_j \mathbf{c}_k + \sum_j \mu_j H_j \mathbf{c}_k \tag{37}$$

$$= \sum_{j \neq k} \mu_j H_j (\mathbf{c}_j - \mathbf{c}_k) + M \mathbf{c}_k \tag{38}$$

Using Eqn. 36, we know that $\mathbf{x} = M^{-1} \sum_j \mu_j H_j \mathbf{c}_j = \mathbf{c}_k + M^{-1} \sum_{j \neq k} \mu_j H_j (\mathbf{c}_j - \mathbf{c}_k)$.

For $M = 2$, we have $\mathbf{x} = \mathbf{c}_2 + M^{-1} \mu_1 H_1 (\mathbf{c}_1 - \mathbf{c}_2)$. So we have

$$(\mathbf{c}_1 - \mathbf{c}_2)^\top H_1 \mathbf{x} = (\mathbf{c}_1 - \mathbf{c}_2)^\top H_1 \mathbf{c}_2 + (\mathbf{c}_1 - \mathbf{c}_2)^\top H_1 M^{-1} H_1 (\mathbf{c}_1 - \mathbf{c}_2) \tag{39}$$

$$\geq (\mathbf{c}_1 - \mathbf{c}_2)^\top H_1 \mathbf{c}_2 \tag{40}$$

This is because $(\mathbf{c}_1 - \mathbf{c}_2) H_1 M^{-1} H_1 (\mathbf{c}_1 - \mathbf{c}_2) \geq 0$ since $H_1 M^{-1} H_1$ is a PSD matrix. Therefore, let $\mathbf{w}_2 := H_1(\mathbf{c}_1 - \mathbf{c}_2)$ and we have $\mathbf{w}_2^\top (\mathbf{x} - \mathbf{c}_2) \geq 0$, which is independent of $\mu \in \Delta$. This means it holds for any $\mathbf{x} \in \Omega_P$.

Let $\mathbf{w}_1 = H_2(\mathbf{c}_2 - \mathbf{c}_1)$, then similarly we have $\mathbf{w}_1^\top (\mathbf{x} - \mathbf{c}_1) \geq 0$ for all $\mathbf{x} \in \Omega_P$. $\square$

## B    QUALITY INDICATORS COMPARISON

Table 1: The review of different scalarizing methods.

| Quality Indicator | Convergence | Uniformity | Spread | No reference set required |
|---|---|---|---|---|
| HyperVolume | $\sqrt{}$ | $\sqrt{}$ | $\sqrt{}$ | $\sqrt{}$ |
| GD | $\sqrt{}$ | | | |
| IGD | $\sqrt{}$ | $\sqrt{}$ | $\sqrt{}$ | |
| MS | | | $\sqrt{}$ | |
| S | | $\sqrt{}$ | | |
| ONVGR | $\sqrt{}$ | | | |

Generational Distance(GD) (Van Veldhuizen & Lamont, 1998) measures the distance the pareto frontier of approximation samples and true pareto frontier, which requires prior knowledge of true pareto frontier and only convergence is considered. IGD (Bosman & Thierens, 2003) is improved version of GD. IGD calculates the distance the points on true pareto frontier to the closest point on pareto frontier of current samples. Inverted Generational Distance(IGD) satisfies all three evaluation metrics of QI but requires true pareto frontier which is hardly to get in real-world problem. Maximum Spread(MS) (Zitzler et al., 2000) computes the distance between the farthest two points of samples to evaluate the spread. Spacing(S) (Bandyopadhyay et al., 2004) measures how close the distribution of pareto frontier of samples is to uniform distribution. Overall Non-dominated Vector Generation and Ratio(ONVGR) is the ratio of number of samples in true pareto-frontier. The table 1 demonstrates the good characteristics of each quality indicators.

## C    END-TO-END LAMOO PSEUDOCODE

Below we list the pseudocode for the end-to-end workflow of LaMOO in Algorithm 2. Specfically, it includes search space partition in **Function Split**. Node(promising region) selection in **Function Select**, and new samples generation in **Function Sample**.

---
**Algorithm 2** LaMOO Pseudocode.

---
1: **Inputs:** Initial $D_0$ from uniform sampling, sample budget $T$.
2: **for** $t = 0, \ldots, T$ **do**
3:     Set $\mathcal{L} \leftarrow \{\Omega_{\text{root}}\}$ (collections of regions to be split).
4:     $\mathcal{V}, v, n \leftarrow \textbf{Split}(\mathcal{L}, D_t)$
5:     $k \leftarrow \textbf{Select}(\mathcal{C}_p, D_t)$
6:     $D_{t+1} \leftarrow \textbf{Sample}(k)$
7: **end for**
8:
9: **Function Split**$(\mathcal{L}, D_t)$
10:     **while** $\mathcal{L} \neq \emptyset$ **do**
11:         $\Omega_j \leftarrow \text{pop\_first\_element}(\mathcal{L})$,  $D_{t,j} \leftarrow D_t \cap \Omega_j$,  $n_{t,j} \leftarrow |D_{t,j}|$.
12:         Compute dominance number $o_{t,j}$ of $D_{t,j}$ using Eqn. 5 and train SVM model $h(\cdot)$.
13:         **If** $(D_{t,j}, o_{t,j})$ is splittable by SVM, **then** $\mathcal{L} \leftarrow \mathcal{L} \cup \text{Partition}(\Omega_j, h(\cdot))$.
14:     **end while**
15:
16: **Function Select**$(C_p, D_t)$
17:     **for** $k = \text{root}$, $k$ is not leaf node **do**
18:         $D_{t,k} \leftarrow D_t \cap \Omega_k$,  $v_{t,k} \leftarrow \text{HyperVolume}(D_{t,k})$,  $n_{t,k} \leftarrow |D_{t,k}|$.
19:         $k \leftarrow \arg \max\limits_{c \,\in\, \text{children}(k)} \text{UCB}_{t,c}$, where $\text{UCB}_{t,c} := v_{t,c} + 2C_p\sqrt{\frac{2\log(n_{t,k})}{n_{t,c}}}$
20:     **end for**
21:     **return** $k$
22:
23: **Function Sample**$(k)$
24:     $D_{t+1} \leftarrow D_t \cup D_{\text{new}}$, where $D_{\text{new}}$ is drawn from $\Omega_k$ based on qEHVI or CMA-ES.
25:     **return** $D_t \cup D_{new}$

---

# D EXPLORATION FACTOR($C_p$) SETUP WITH UNKNOWN MAXIMUM HYPERVOLUME

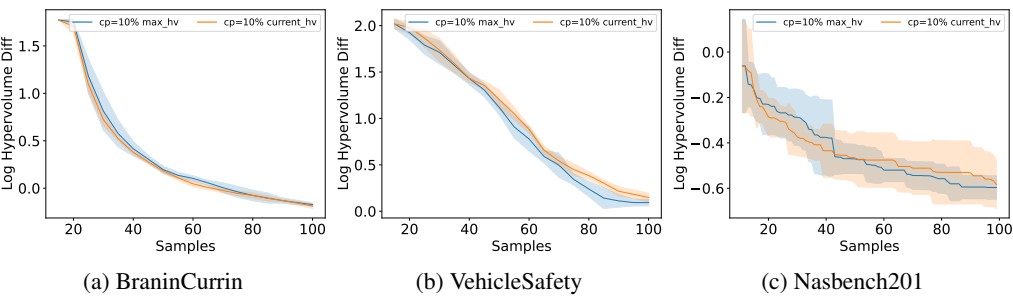

Figure 9: Sampling with static $C_p$(10% of $HV_{max}$) and dynamic $C_p$((10% of $HV_{current}$))

As we mentioned in the paper, a "rule of thumb" is to set the $C_p$ to be roughly 10% of the maximum hypervolume HVmax. If HVmax is unknown, $C_p$ can be dynamically set to 10% of the hypervolume of current samples in each search iteration. The figures below demonstrate the difference between 10% HVmax and 10% current hypervolume in three problems(Branin-Currin, VehicleSafety, and Nasbench201 from left to right). The final performances by using 10% HVmax and 10% current hypervolume are similar.

# E WALL CLOCK TIME IN DIFFERENT PROBLEMS

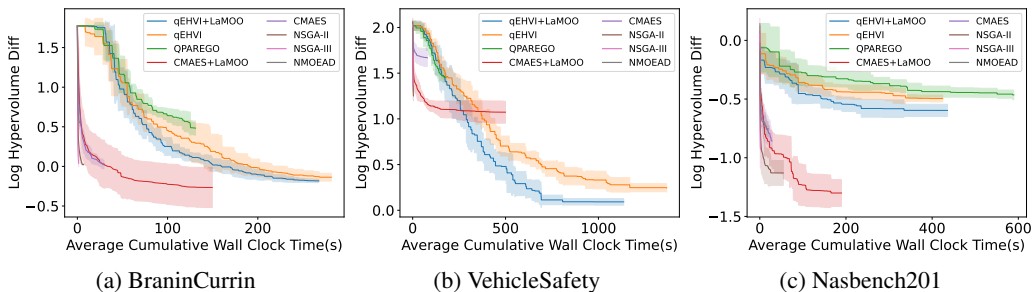

Figure 10: Wall clock time in different problems

Fig. 10 shows the wall clock time of different search algorithms in BraninCurrin(Belakaria et al., 2019), VehicleSafefy (Liao et al., 2008) and Nasbench201 (Dong & Yang, 2020).

# F DETAILS OF BENCHMARK PROBLEMS

## F.1 PROBLEM DESCRIPTION

**BraninCurrin (Belakaria et al., 2019)**:

$$f^{(1)}(x_1, x_2) = (15x_2 - \frac{5.1 * (15x_1 - 5)^2}{4\pi^2} + \frac{75x_1 - 25}{\pi} - 5)^2 + (10 - \frac{10}{8\pi}) * \cos(15x_1 - 5)$$
$$f^{(2)}(x_1, x_2) = \left[1 - \exp\left(-\frac{1}{(2x_2)}\right)\right]\frac{2300x_1^3 + 1900x_1^2 + 2092x_1 + 60}{100x_1^3 + 500x_1^2 + 4x_1 + 20}$$

where $x_1, x_2 \in [0, 1]$.

**VehicleSafefy (Liao et al., 2008):**

$$f_1(\boldsymbol{x}) = 1640.2823 + 2.3573285x_1 + 2.3220035x_2 + 4.5688768x_3 + 7.7213633x_4 + 4.4559504x_5$$
$$f_2(\boldsymbol{x}) = 6.5856 + 1.15x_1 - 1.0427x_2 + 0.9738x_3 + 0.8364x_4 - 0.3695x_1x_4 + 0.0861x_1x_5$$
$$\qquad + 0.3628x_2x_4 + 0.1106x_1^2 - 0.3437x_3^2 + 0.1764x_4^2$$
$$f_3(\boldsymbol{x}) = -0.0551 + 0.0181x_1 + 0.1024x_2 + 0.0421x_3 - 0.0073x_1x_2 + 0.024x_2x_3 - 0.0118x_2x_4$$
$$\qquad - 0.0204x_3x_4 - 0.008x_3x_5 - 0.0241x_2^2 + 0.0109x_4^2$$

where $\boldsymbol{x} \in [1, 3]^5$.

**Nasbench201 (Dong & Yang, 2020):**

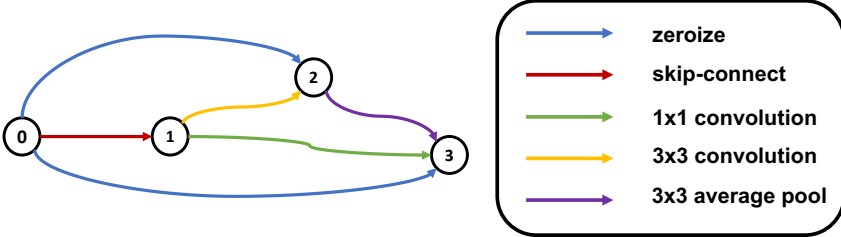

Figure 11: A general architecture of Nasbench201

In Nasbench201, the architectures are made by stacking the cells together. The difference among architectures in Nasbench201 is the design of the cells, see fig 11. Specifically, each cell contains 4 nodes, and there is a particular operation connecting to two nodes including zeroize, skip-connect, 1x1 convolution, 3x3 convolution, and 3x3 average pooling. Therefore, there are $C_4^2 = 6$ edges in a cell and $5^6 = 15625$ unique architectures in Nasbench201. According to this background, Each architecture can be encoded into a 6-dimensional vector with 5 discrete numbers (i.e., 0, 1, 2, 3, 4 that corresponds to zeroize, skip-connect, 1x1 convolution, 3x3 convolution, and 3x3 average pooling).

$$f_1(\boldsymbol{x}) = Accuracy(\boldsymbol{x})$$
$$f_2(\boldsymbol{x}) = \#FLOPs(\boldsymbol{x})$$

where $\boldsymbol{x} \in \{0, 1, 2, 3, 4\}^6$.

**DTLZ2 (Deb et al., 2002b):**

$$f_1(\boldsymbol{x}) = (1 + g(\boldsymbol{x}_M)) \cos\left(\frac{\pi}{2}x_1\right) \cdots \cos\left(\frac{\pi}{2}x_{M-2}\right) \cos\left(\frac{\pi}{2}x_{M-1}\right)$$
$$f_2(\boldsymbol{x}) = (1 + g(\boldsymbol{x}_M)) \cos\left(\frac{\pi}{2}x_1\right) \cdots \cos\left(\frac{\pi}{2}x_{M-2}\right) \sin\left(\frac{\pi}{2}x_{M-1}\right)$$
$$f_3(\boldsymbol{x}) = (1 + g(\boldsymbol{x}_M)) \cos\left(\frac{\pi}{2}x_1\right) \cdots \sin\left(\frac{\pi}{2}x_{M-2}\right)$$
$$\vdots$$
$$f_M(\boldsymbol{x}) = (1 + g(\boldsymbol{x}_M)) \sin\left(\frac{\pi}{2}x_1\right)$$

where $g(\boldsymbol{x}) = \sum_{x_i \in \boldsymbol{x}_M}(x_i - 0.5)^2$, $\boldsymbol{x} \in [0, 1]^d$, and $\boldsymbol{x}_M$ represents the last $d - M + 1$ elements of $\boldsymbol{x}$.

## F.2 Visualization of Pareto-Frontier for Benchmark Problems

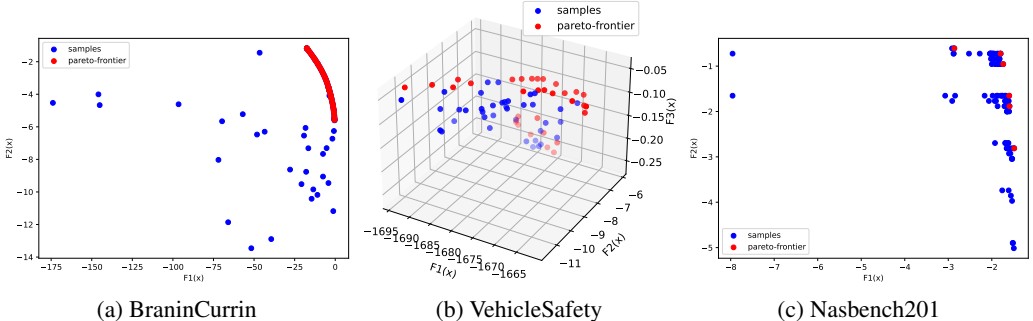

(a) BraninCurrin      (b) VehicleSafety      (c) Nasbench201

Figure 12: Visualization of Pareto-frontier in BraninCurrin, VehicleSafety as well as Nasbench201.

## F.3 Reference Points

$^{\dagger}$: $M$ represents the number of objectives.

| Problem | Reference Point |
|---|---|
| BraninCurrin | (18.0, 6.0) |
| VehicleSafety | (1864.72022, 11.81993945, 0.2903999384) |
| Nasbench201 | (-3.0, -6.0) |
| DTLZ2 | $(1.1, .., 1.1) \in \mathbb{R}^{\mathbb{M}^{\dagger}}$ |
| Molucule Discovery | $(0.0, ..., 0.0) \in \mathbb{R}^{\mathbb{M}^{\dagger}}$ |

Table 2: The reference points for all problems in this work.

The reference point $R \in \mathbb{R}^M$ is defined to measure the hypervolume of a problem. Different reference point would result in a different hypervolume. The details can be found at sec 1. Table 2 elaborates the reference points in the problems throughout the paper.

## F.4 Maximum Hypervolume of each problem

| Problem | Maximum Hypervolume |
|---|---|
| BraninCurrin | 59.36011874867746 |
| VehicleSafety | 246.81607081187002 |
| Nasbench201 | 8.06987476348877 |
| DTLZ2(2 objectives) | 1.4460165933151778 |
| DTLZ2(10 objectives) | 2.5912520655298095 |
| Molucule Discovery | N/A |

Table 3: The maximum hypervolume for all problems in this work.

Table 3 elaborates the observed maximal hypervolume in the problems throughout the paper. We used these value to calculate the log hypervolume difference in fig 3 and fig 4.

## G Experiments Setup

Experiment details: For small-scale problems(i.e. Branin-Currin, VehicleSafety, and Nasbench201) and DTLZ2 with 2 and 10 objectives. We randomly generate 10 samples as the initialization. For

multi-objective molecule discovery, the number of initial samples is 150. In each iteration, we update 5 batched samples(q value) for all search algorithms.

Hyperparameters of LAMOO: For all problems, we leverage polynomials as the kernel type of SVM and the degree of the polynomial kernel function is set to 4. The minimum samples in the leaf of MCTS is 10. The cp is roughly set to 10% of maximum of hypervolume(i.e. Branin-Currin -> 5, VehicleSafety -> 20, Nasbench201 -> 6, DTLZ2(2 objectives) -> 0.1, DTLZ2(10 objectives) -> 0.25, molecule discovery(2 objectives) -> 0.03, molecule discovery(3 objectives) -> 0.2, molecule discovery(4 objectives) -> 0.06).

Hyperparameters of qEHVI and qParEGO: The number of q is set to 5. The acquisition function is optimized with L-BFGS-B (with a maximum of 200 iterations). In each iteration, 256 raw samples used for the initialization heuristic are generated to be selected by the acquisition function. In original work Daulton et al. (2020), they used 1024 raw samples but we decrease this number to 256 to sample budget of all methods for comparison, which speeds up the search but may lead to lower performance such as vehiclesafty problem in fig. 3. As the same claim in Daulton et al. (2020), each generated sample is modeled with an independent Gaussian process with a Matern 5/2 ARD kernel.

## H    VERIFICATION OF LAMOO ON MANY-OBJECTIVE PROBLEMS

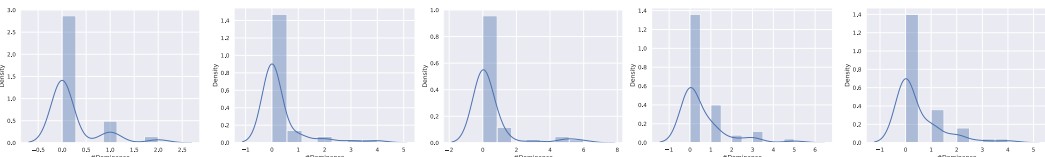

Figure 13: Dominance number distribution with 50 random samples on DTLZ2(10 objectives)

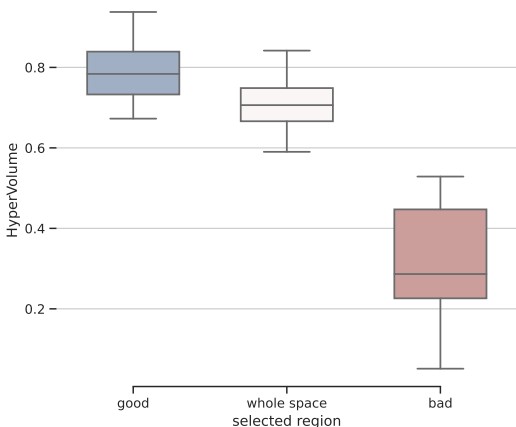

Figure 14: The range of hypervolume for 50 samples randomly generated from different regions in DTLZ2(10 objectives). We generate 25 times of 50 samples in total.

While it is theoretically hard to label the samples into good and bad based on their dominance number in many-objective problems due the the lack of dominance pressure(All samples are non-dominated with each other). If number of objective is not too large(i.e. $M \leq 10$), the samples can be still split by dominance number. Given the problem(DTLZ2 with 10 objectives) shown in fig 4, we randomly generate 50 samples in the search space and draw the dominance distribution of them(see fig 13). We did this experiment 5 times.

We then partition the search space by a SVM classifier based on the labeled samples into "good" and "bad", and randomly generate 50 samples in "good region", "bad region", and whole space, respectively. We did this process 5 times with different initial samples. Fig. 14 shows the range of hypervolume of the samples generated from good regions, the whole space, and bad regions. From

the figure, we can see that the hypervolume of samples generated from good regions are significantly higher than others.

# I    COMPUTATIONAL COMPLEXITY ANALYSIS OF LAMOO

Here is a detailed breakdown of the computational complexity of our algorithm (Alg. 1)

Line.6: Compute dominance: $O(MN_{node}^2)$ where $N_{node}$ is the number of samples in the node and $M$ is the number of dimensions.

Line 7: Time complexity of SVM : $O(N_{node}^2)$ where $N_{node}$ is the number of samples in the node.

Line 10: Hypervolume: $O(N^{\frac{M}{2}} + N \log N)(M > 3)$ (Beume & Rudolph, 2006) or $O(N \log N)(M \leq 3)$ (Beume et al., 2009), where $N$ is number of searched samples in total and $D$ is the number of dimensions.

Total time complexity: $\sum_{i=1}^{t} O(N_i^2)(M < 3)$, where $t$ is the total number of nodes. $\sum_{i=1}^{t} O(N^{\frac{M}{2}} + N_i \log N_i)$ $(M > 3)$, where $t$ is the total number of nodes.

When there are more than 3 objectives ($M > 3$), HV computation is the dominant factor. When $M \leq 3$, the optimization cost of SVM is the dominant factor.

# J    VARIATION OF LAMOO WITH A CHEAPER OVERHEAD

---

**Algorithm 3** LaMOO Pseudocode with leaf based selection.

---

1: **Inputs:** Initial $D_0$ from uniform sampling, sample budget $T$.
2: **for** $t = 0, \dots, T$ **do**
3:     Set $\mathcal{L} \leftarrow \{\Omega_{\text{root}}\}$ (collections of regions to be split).
4:     **while** $\mathcal{L} \neq \emptyset$ **do**
5:         $\Omega_j \leftarrow$ pop_first_element($\mathcal{L}$), $D_{t,j} \leftarrow D_t \cap \Omega_j$, $n_{t,j} \leftarrow |D_{t,j}|$.
6:         Compute dominance number $o_{t,j}$ of $D_{t,j}$ using Eqn. 5 and train SVM model $h(\cdot)$.
7:         **If** $(D_{t,j}, o_{t,j})$ is splittable by SVM, **then** $\mathcal{L} \leftarrow \mathcal{L} \cup \text{Partition}(\Omega_j, h(\cdot))$.
8:     **end while**
9:     **for** $k = $ root, $k$ is not leaf node **do**
10:         $D_{t,k} \leftarrow D_t \cap \Omega_k$, $n_{t,k} \leftarrow |D_{t,k}|$.
11:     **end for**
12:     **for** $l$ is leaf node **do**
13:         $v_{t,l} \leftarrow \text{HyperVolume}(D_{t,l})$
14:     **end for**
15:     $k \leftarrow \arg \max_{l \in \text{ leaf nodes}} \text{UCB}_{t,l}$, where $\text{UCB}_{t,l} := v_{t,l} + 2C_p\sqrt{\frac{2 \log(n_{t,l})}{n_{t,p}}}$, where $p$ is the parent of $l$.
16:     $D_{t+1} \leftarrow D_t \cup D_{\text{new}}$, where $D_{\text{new}}$ is drawn from $\Omega_k$ based on qEHVI or CMA-ES.
17: **end for**

---

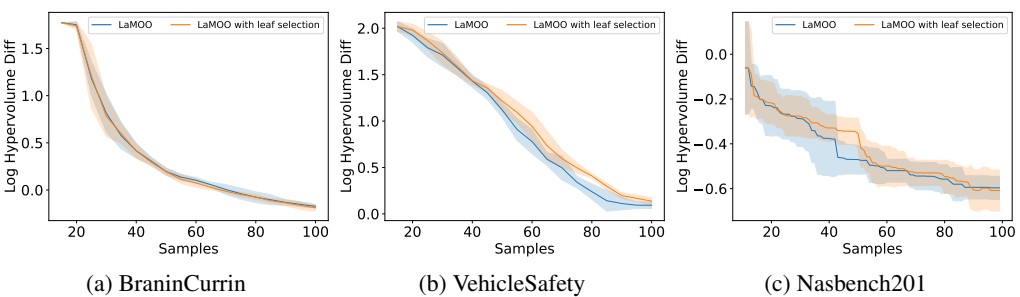

(a) BraninCurrin               (b) VehicleSafety               (c) Nasbench201

Figure 15: Search progress with sample

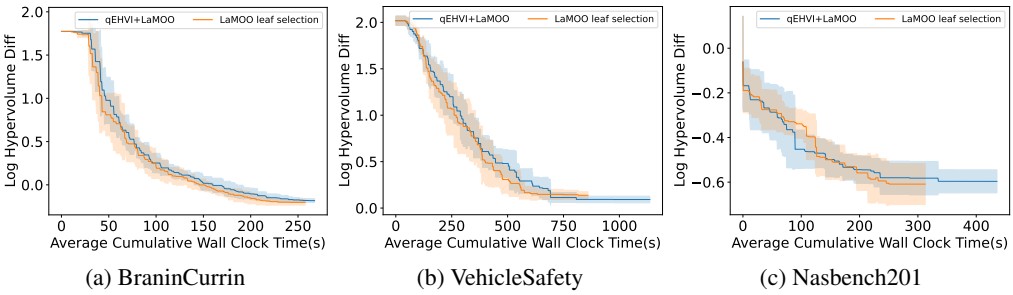

Figure 16: Search progress with time

Instead of traversing down the search tree to trace the current most promising search path, this variation of LaMOO directly select the leaf node with the highest *UCB value*. Algorithm. 3 illustrates the detail of this variation. Therefore, this variation avoids calculating the hypervolume in the non-leaf nodes of the tree, where hypervolume calculation is the main computational cost of LaMOO especially in many-objective problems. Figure. 15 and figure. 16 validate the variation that is able to reach similar performance of searched samples but saves lots of time. We leave the validation of others problems in the future works.

## K   ADDITIONAL RELATED WORKS

(Hashimoto et al., 2018a; Kumar et al., 2018; Hashimoto et al., 2018b) indeed leverage classifiers to partition search space and draw the new samples from the good region. However, (Hashimoto et al., 2018a; Kumar et al., 2018; Hashimoto et al., 2018b) randomly sampled in selected regions without integrating existing optimizers (e.g. Bayesian optimization, evolutionary algorithms). In addition, they progressively select the good regions without a trade-off of exploration and exploitation as we did by leveraging Monte Carlo Tree Search (MCTS). (Munos, 2011a; Wang et al., 2014; Kawaguchi et al., 2015) can be seen as the first work to use MCTS to build hierarchical space partitions. But their partitions are predefined (e.g., Voronoi graph, axis-aligned partition, etc) without learning (or adapting to) observed samples so far, except for (Wang et al., 2019; 2020; Yang et al., 2021), which are learning extensions coupled with MCTS. However, they all deal with single-objective optimization.

For multi-objective optimization, (Loshchilov et al., 2010; Seah et al., 2012; Pan et al., 2019) learns to predict the dominance rank of samples, without computing them algorithmic-ally, a slow process with many previous samples, in order to speed up the MOEAs algorithms. Unlike our paper, they do not partition the search space into good/bad regions. In contrast, LaMOO computes the rank algorithmic-ally. Therefore, our contributions are complementary to theirs. We leave a combination of both as one of the future works.

LaMOO V.S. LaMCTS/LaNAS: First, the mechanism of the partitioning of the search space is different. LaMOO uses dominance rank to separate good from bad regions, while LaMCTS uses a k-mean for region separation. LaNAS is even more simple: it uses the median from the single objectives of currently collected samples and a linear classifier to separate regions.

LaMOO V.S. LaP³: LAP³ is a planning algorithm tailored to RL with a single objective function. LAP³ also utilizes the representation learning for the partition space and planning space, while our LaMOO doesn't.

