# OpenReview forum: "Multi-objective Optimization by Learning Space Partition"
_ICLR.cc/2022/Conference — ICLR 2022 Poster_

### Official Review · Reviewer_Vetj · 2021-11-02

**Correctness:** 3
**Technical Novelty And Significance:** 3
**Empirical Novelty And Significance:** 3
**Recommendation:** 6
**Confidence:** 4

**Main Review:**

This paper seems to be the first work by extending the space-partition approach for SOO to MOO. The metric of dominance number is reasonable and easy-to-follow. This paper is well organized, with good background understanding and proper motivations. Sufficient theoretical analysis and experimental demonstration are provided.
However, this paper still suffers from the issue of insufficient presentation. Some revisions are need to further improve the quality of the paper, listed as:
1.	The full name of UCT should be given.
2.	In Algorithm 1, the sample budget T is used as the termination condition. But it is unclear how to set it in the experiments.
3.	The computational overhead of LaMOO is not very clear. It is better to give a time complexity analysis for LaMOO.
4.	In LaMOO with CMA-ES, the generated solution is constrained to be within the leaf region, but how? In addition, if such a constrain is frequently implemented, does it deteriorate the search efficiency?
5.	It is unclear why the optimization dimension of Nasbench201 is 6. More interpretations are needed.
6.	In page 8, the authors said there are 1000 samples given for three different tasks. But in Fig. 4, only 200 samples are shown. Why?
7.	How to set Cp when HV_max is unknown?
8.	Please have your submission proof-read for English style and grammar issues. For example, there are syntax errors in “LaMOO+qEHVI achieve”, “Thanks to MCTS, LaMOO also considers …are given 1000 samples”, “LaMOO help”, “to performs”, just mention a few.
9.	The paper “Learning space partitions for path planning” was cited twice.


**Summary Of The Paper:**

This paper presents a learning space partitions-based multi-objective optimization framework by using Monte Carlo tree search and an innovatively proposed metric, i.e., dominance number. Solid theoretical analysis on single-objective optimization (SOO) and some observation on multi-objective optimization (MOO) for the space partitions are given. Extensive experiments are conducted based on synthetic functions, vehicle safety problem, Nasbench201 neural architecture search problem, and molecular design. Comparative results with other baselines show the superiority and effectiveness of the proposed LaMOO.

**Summary Of The Review:**

A good paper with clear motivation, interesting idea, solid theoretical and experimental demonstration, but some presentations are not very clear.

---

> ### Author Response · Authors · 2021-11-19
> **Response to Reviewer Vetj**
>
> We sincerely thank the reviewer for your insightful comments and suggestions, please check our answers below.
>
> ---
> **The full name of UCT should be given**
>
> We added the full name of UCT in our next revision.
>
> ---
> **how to set T in the experiments**
>
> In practice, if the hypervolume of searched samples is not increased in several iterations, the search would be terminated.
>
> ---
> **Time complexity analysis for LaMOO**
>
> Please check the answers to common questions(**Computational complexity & computational overhead of LaMOO**). Thank you.
>
> ---
> **Search efficiency of LaMOO with CMA-ES**
>
> The system would rule out the generated solutions that are beyond the selected region and re-generate new samples until they satisfy the constraints. Yes, it is indeed frequently encountered but generating new solutions by CMA-ES is very cheap, so it does not significantly deteriorate the search efficiency. We have reported the wall clock time for each search algorithm in **figure.10 in Appendix. E**, from the figure, while LAMOO with CMA-ES costs more time compared to original CMA-ES in the same sample budget, LAMOO with CMA-ES outperforms original CMA-ES by using the same search time.  Also, if a function evaluation takes a long time (as is often the case for multi-objective optimization), then this overhead will be relatively minor.
>
> ---
> **Unclear definition of optimization dimension of Nasbench201**
>
> We added the description of the optimization dimension of Nasbench201 in **Appendix.F1**. I would also support a description here. In Nasbench201, the architectures are made by stacking the cells together. The difference among architectures in Nasbench201 is the design of the cells. Specifically, each cell contains 4 nodes, and there is a particular operation connecting to two nodes including zeroize, skip-connect, 1x1 convolution, 3x3 convolution, and 3x3 average pooling. Therefore, there are $C_{4}^{2}$ = **6** edges in a cell and $5^{6}$ =15625 unique architectures in Nasbench201. According to this background, Each architecture can be encoded into a 6-dimensional vector with 5 discrete numbers (i.e., 0, 1, 2, 3, 4 that corresponds to zeroize, skip-connect, 1x1 convolution, 3x3 convolution, and 3x3 average pooling).
>
> ---
> **In page 8, the authors said there are 1000 samples given for three different tasks. But in Fig. 4, only 200 samples are shown. Why?**
>
> Sorry for this confusion, we searched 1000 samples by evolutionary-based algorithms for small-scale tasks **(Branin-Currin, VehicleSafety, and  Nasbench201)** as shown in **Fig.3**. For many-objective optimization problems, it is hard to calculate hypervolume with a large size of samples. Therefore, we only search the many-objective optimization problem in 200 samples in **Fig.4**.
>
> ---
>
> **How to set Cp when HV_max is unknown?**
>
> Please check the answers to common questions(**Guidelines to set the exploration factor C_p**). Thanks.
>
> ---
> **Grammar issues.**
>
> Thanks for pointing the grammar issues in the paper. We have fixed all the issues from you and fully polished our paper.
>
> ---
> **Twice cited "Learning space partitions for path planning”**
>
> Thank you for pointing it out. It has been fixed.

---

> > ### Comment · Reviewer_Vetj · 2021-11-26
> > **Comments Added After Reading Author Response**
> >
> > Authors have addressed all the issues according to my previous comments.

---

### Official Review · Reviewer_2MfT · 2021-11-02

**Correctness:** 3
**Technical Novelty And Significance:** 3
**Empirical Novelty And Significance:** 3
**Recommendation:** 6
**Confidence:** 4

**Main Review:**

**Strengths:**

+ This work is well-organized and generally well-written.

+ The studied problem, multi-objective optimization, is important and could be useful for many real-world applications.

+ The proposed LaMOO method is a reasonable generalization from LaMCTS to multi-objective optimization.

**Weakness:**

*Method:*

**1. Novelty:**

- **Incremental Contribution:** The proposed LaMOO is a direct generalization from the LaMCTS method to multi-objective optimization (MOO). The novel part is to use dominance number as criteria for search space partition and hypervolume for promising region selection. These are all straightforward generalizations for MOO. The contribution of this work is somewhat incremental along the line of LaNAS, LaMCTS, and LaP^3.

- **Missing Closely Related Approaches:** This work claims the proposed approach to learn the promising region is *fundamentally different* from the previous works. However, many classification-based search space partition methods have been proposed in the machine learning community, see [1][2][3] (classification + random sampling). (Tree-based) space partition methods have been widely used for black-box optimization [4][5][6]. In addition, there are also different works on classification-based MOO [7] (SVM + NSGA-II/MO-CMA-ES) [8] (Ordinal SVM + NSGA-II) [9].


**2. Theoretical Analysis:**

A large part of this work is on the theoretical understanding for space partition and LaMCTS. However, the analysis is mostly for single-objective optimization, and the extension to multi-objective optimization is much less promising.

**3. Why LaMOO Works:**

Further discussions are needed to clearly clarify the properties of LaMOO.

- **Dominance-based Approach for Many Objective Optimization:** LaMOO uses the dominance number as the split criteria to train the SVM models and partition the search space. However, the dominance-based method is typically not good for many objective optimization due to the lack of dominance pressure (e.g., all solutions are non-dominated with each other, and all have the same dominated number). Why is LaMOO still good for many objective optimization?

- **Combination with Multi-Objective Bayesian Optimization (MOBO):** It is straightforward to see the benefit of using LaMOO with model-free optimization (e.g., NSGA-II and MO-CMA-ES). However, it is not so clear to understand why it also works for MOBO (e.g., qEHVI). The qEHVI approach already builds (global) Gaussian process models to approximate each objective function, and uses hypervolume-based criteria to select the most promising solution(s) (e.g., maximizing the expected hypervolume improvement) for evaluation. Therefore, its selected solution(s) should be already on the approximate Pareto front without the LaMOO approach.  Is the good performance due to only use solutions in the promising region to build the models (but I think GP would work well with all data as in the setting considered in this work)? Or because LaMOO restricts the search in the region close to the current best non-dominated solutions (then what is the relation to the trust-region approach [10])?

- **Exploitation v.s. Exploration:** With LaMOO, the solutions can only be selected from the most promising region (e.g., around the current Pareto front), which is good for exploitation. However, will this approach lead to worse overall performance due to the lack of exploration (e.g., cannot find more diverse Pareto solutions far from the current Pareto front)?

**4. Time Complexity:**

What is the time complexity of the proposed algorithm? In each step of LaMOO, it has to repeatedly calculate the hypervolume of different regions for promising region selection. However, the computation of hypervolume could be time-consuming, especially for problems with many objectives (e.g., >3). Would it make LaMOO impractical for those problems?

**5. Inaccurate Description for MOO Methods:**

- **CMA-ES:** CMA-ES is a widely-used single objective optimization algorithm [11]. The multi-objective version proposed in (Igel et al., 2007a) is usually called MO-CMA-ES. It is also confusing why most citation for the MO-CMA-ES (in the main paper and Table 1) is for the steady-state updated version (Igel et al., 2007b) but not for the original paper (Igel et al., 2007a).

- **ParEGO:** The seminal algorithm proposed in Knowles (2006) is called ParEGO and the qParEGO is a parallel extension recently proposed in Daulton et al. (2020). It is not suitable to refer the algorithm in Knowles (2006) as qParEGO in Table 1 and the main text.

- **MOEA/D:** In my understanding, MOEA/D is suitable for many objective optimization (objectives > 3), see its performance in the NSGA-III paper (Deb & Jain, 2014), while the main challenge is how to specify the weight vector for a new problem with unknown Pareto front as correctly pointed out in this work.

- **Hypervolume-based Method:** This work indicates the indicator-based method is better for many objective optimization. However, the time complexity and expensive calculation could make the hypervolume-based method impractical for many-objective optimization.

*Experiment:*

**6. Missing Experimental Setting:**

Many important experiment settings are missing in this work, such as the number of initial solutions for MOBO (and its generation method), the number of batched solutions for MOBO (e.g., q), the reference point for hypervolume (during the optimization, and for the final evaluation), the ground truth Pareto front used for calculating the log hypervolume difference for real-world problems (e.g., Nasbench 201).

**7. Comparison to Model-Free Evolutionary Algorithm:**

It is reasonable that LaMOO can improve the MO-CMA-ES performance since it builds extra models to allocate computation to the most promising region. However, in my understanding, the model-free evolutionary algorithms are not designed for expensive optimization, and their typical use case is with a large number of cheap evaluations with a fast run time. It is more interesting to directly compare LaMOO with other model-based methods (e.g., MO-CMA-ES with GP models).

**8. MOBO Performance:**

What are the hyperparameters for qEHVI? It seems its performance on VehicleSafty problem is worse than those reported in the original paper Daulton et al. (2020).

**9. Wall Clock Run Time:**

Please report the wall clock run time for both LaMOO and other model-free/model-based algorithms, as in Daulton et al. (2020).


**Minor Issues:**

When citing multiple works, please put them in chronological order.

**Reference:**

[1] Hashimoto, Tatsunori, Steve Yadlowsky, and John Duchi. Derivative free optimization via repeated classification. AISTATS 2018.

[2] Kumar, Manoj, George E. Dahl, Vijay Vasudevan, and Mohammad Norouzi. Parallel architecture and hyperparameter search via successive halving and classification. arXiv:1805.10255.

[3] Yu, Yang, Hong Qian, and Yi-Qi Hu. Derivative-free optimization via classification. AAAI 2016.

[4] Munos, Rémi. Optimistic optimization of a deterministic function without the knowledge of its smoothness. NeurIPS 2011.

[5] Ziyu Wang, Babak Shakibi, Lin Jin, and Nando de Freitas. Bayesian multi-scale optimistic optimization. AISTATS 2014.

[6] Kenji Kawaguchi, Leslie Pack Kaelbling, and Tomas Lozano-Perez. Bayesian optimization with exponential convergence. NeurIPS 2015.

[7] Loshchilov, Ilya, Marc Schoenauer, and Michèle Sebag. A mono surrogate for multiobjective optimization. In Proceedings of the 12th annual conference on Genetic and evolutionary computation, 2010.

[8] Seah, Chun-Wei, Yew-Soon Ong, Ivor W. Tsang, and Siwei Jiang. Pareto rank learning in multi-objective evolutionary algorithms. In 2012 IEEE Congress on Evolutionary Computation, 2012.

[9] Pan, Linqiang, Cheng He, Ye Tian, Handing Wang, Xingyi Zhang, and Yaochu Jin. A classification-based surrogate-assisted evolutionary algorithm for expensive many-objective optimization. IEEE Transactions on Evolutionary Computation 2018.

[10] Daulton, Samuel, David Eriksson, Maximilian Balandat, and Eytan Bakshy. Multi-Objective Bayesian Optimization over High-Dimensional Search Spaces. arXiv:2109.10964, 2021.

[11] Hansen, Nikolaus, and Andreas Ostermeier. Completely derandomized self-adaptation in evolution strategies. Evolutionary Computation 2001.

[12] Ishibuchi, Hisao, Yu Setoguchi, Hiroyuki Masuda, and Yusuke Nojima. Performance of decomposition-based many-objective algorithms strongly depends on Pareto front shapes. IEEE Transactions on Evolutionary Computation 21, no. 2 (2016): 169-190.





**Summary Of The Paper:**

This work proposes a novel learning-based method called  LaMOO to partition the search space for the multi-objective optimization problem. With the learned partition, the computational budget can be allocated to the small promising regions (e.g., the region close to the Pareto frontier) rather than the whole search space. It is a direct generalization from the closely related works (e.g., LaNAS, LaMCTS, and LaP^3) to multi-objective optimization. The proposed LaMOO has promising experimental results on both synthetic functions and real-world optimization problems.

**Summary Of The Review:**

This work tackles an important research problem that could be useful for different partitioners in the community (e.g., multi-objective optimization, Bayesian optimization, and NAS). The proposed method is a reasonable generalization from the LaMCTS method. However, due to the concerns listed above on both the method and the experimental results, I cannot give a clear acceptance to the current manuscript.

---

> ### Author Response · Authors · 2021-11-19
> **Response to Reviewer 2MfT (1/3)**
>
> We sincerely thank the reviewer for your insightful comments and suggestions, please check our answers below.
>
> ---
> **1. Novelty:**
>
> The reviewer claims that the paper has **incremental contribution**.
>
> We kindly disagree with the reviewer's claim and emphasize that LaMOO proposes a nontrivial contribution compared to existing works (LaMCTS/LaNAS/LaP^3). While LaMOO leverages a similar idea (which is "learn to partition"), it addresses multi-objective optimization (MOO), a very different research problem than single-objective optimization (SOO). None of LaMCTS/LaNAS/LaP^3 have addressed MOO yet.
>
> It would be awkward to say that approaches for MOO is an "simple* extension of approaches for SOO, given that there is already a community working on MOO for decades. In particular, the reviewer has already listed a few MOO papers and emphasized their difference compared to their SOO counterpart (e.g., CMA-ES versus MO-CMA-ES).
>
> Besides, LaMOO contains important difference shown below:
>
> **LaMOO V.S. LaMCTS/LaNAS**: First, the mechanism of the partitioning of the search space is different. LaMOO uses dominance rank  to separate good from bad regions, while LaMCTS uses a k-mean for region separation. LaNAS is even more simple: it uses the median from the single objectives of currently collected samples and a linear classifier to separate regions.
>
> **LaMOO V.S. LaP^3**: LAP^3 is a planning algorithm tailored to RL with a single objective function. LAP^3 also utilizes the representation learning for the partition space and planning space, while our LaMOO doesn't.
>
> ---
> **Missing Closely Related Works**
>
> [1][2][3] indeed leverage classifiers to partition search space and draw the new samples from the good region. However, [1][2][3] randomly sampled in selected regions without integrating existing optimizers (e.g. Bayesian optimization, evolutionary algorithms). In addition, they progressively select the good regions without a trade-off of exploration and exploitation as we did by leveraging Monte Carlo Tree Search (MCTS). [4][5][6]can be seen as the first work to use MCTS to build hierarchical space partitions. But their partitions are predefined (e.g., Voronoi graph, axis-aligned partition, etc) without learning (or adapting to) observed samples so far, except for LA-MCTS / LaNAS / LaP^3, which are learning extensions coupled with MCTS. However, they all deal with single-objective optimization.
>
> For multi-objective optimization, [7][8][9] learns to predict the dominance rank of samples, without computing them algorithmically, a slow process with many previous samples, in order to speed up the MOEAs algorithms. Unlike our paper, they do not partition the search space into good/bad regions. In contrast, LaMOO computes the rank algorithmically. Therefore, our contributions are complementary to theirs. We leave a combination of both as one of the future works.
>
> While LaMOO shows that learning-to-partition empirically works well in multi-objective optimization, we acknowledge that our work is built on top of many existing works and will tune down our claim a bit. For example, we will remove the word “fundamental” in the introduction add these related works in our next revision(appendix.K).
>
> ---
> **2.Theoretical Analysis**
>
> While we acknowledge that our theoretical analysis (Sec. 3) is mainly on single-objective optimization, we indeed have a section (**Sec. 3.3**) that demonstrates that in certain multi-objective optimization problems, Pareto optimal sets can be separated from nonoptimal sets by linear boundaries, which demonstrates that our technique should work in these cases. We leave a formal theoretical analysis on multi-objective optimization as the future work.

---

> ### Author Response · Authors · 2021-11-19
> **Response to Reviewer 2MfT (2/3)**
>
> ---
> **3. Why LaMOO Works**
>
> **Dominance-based Approach for Many Objective Optimization**
>
> Thanks for pointing out this potential limitation of LaMOO. Multi-objective optimization with many objectives is typically hard, as shown in many[r-1][r-2][r-3]. We agree that there exist situations where the samples are scattered around in multiple directions and all of them can be part of the frontier, and space partitions may not work.
>
> However, in practice, for many real-world problems, good/bad regions can be immediately clear after a decent amount of samples are collected, and good regions (i.e., good in many/most objectives) often reside in a small place.
>
> We added a section in **appendix.H** to show how LaMOO works on this many-objective problem. We would also write down my reasons below.
>
> Given the problem(DTLZ2 with 10 objectives) shown in our paper, we randomly generate 50 samples in the search space and draw the dominance distribution of them(see **Figure.13 in appendix H**). We did this experiment 5 times.
>
> The **figure.13** shows that 50 samples can still be split into ‘good’ and ‘bad’ in 10 objective problems. We then partition the search space by an SVM classifier based on the labeled samples into “good” and “bad” and randomly generate 50 samples in "good region", “bad region”, and whole space, respectively. We did this process 5 times for each classifier with different initial samples. **Figure.14** shows the range of hypervolumes of the samples generated from the good region, the whole space, and the bad region.  From the figure, we can see that the hypervolume of samples generated from good regions is significantly higher than others.
>
> ---
> **Combination with Multi-Objective Bayesian Optimization (MOBO)**
>
> Solving the problem on a search space partition is simpler than solving the entire search space using MOBO. For example, solving the acquisition in MOBO is a lot easier with LaMOO for only the selected partition. But the regular MOBO needs to solve the acquisition on the entire search space, which is the main bottleneck in practice.
>
> ---
> **Exploitation v.s. Exploration**
>
> The reviewer may have misunderstood our algorithm (**Alg. 1**). LaMOO keeps both the most promising and less promising regions and strikes a balance among them, by plugging in MCTS (and UCB score) to balance exploitation and exploration. Setting a good exploration factor(C_P) can control the importance of exploration. Please check “**the exploration factor Cp**” in Ablation of Design Choices to get more details. While our theoretical analysis indeed only considers greedy cases, our empirical algorithm (Alg. 1) takes this trade-off into consideration.
>
> ---
> **4. Time Complexity**
>
> Please check the answer to common questions(**Computational complexity & computational overhead of LaMOO**). Thanks.
>
>
> ---
> **5. Inaccurate Description for MOO Methods**
>
> We thank the reviewer for the detailed comments and revised the paper accordingly.
>
> ---
> **6. Missing Experimental Setting**
>
> Thanks for the useful suggestion.
>
> We added the description of problems and experiment setup in **appendix.F** and **appendix.G**.
>
> ---
> **7. Comparison to Model-Free Evolutionary Algorithm**
>
> The reviewer might misunderstand our proposed algorithm, we have compared both model-free and model-based baselines. We have integrated Bayesian Optimization and EA under the LaMOO framework to show that our method works with both model-free (and cheap evaluations) and model-based (expensive evaluations). These should cover two very different scenarios. This integration has demonstrated LaMOO is an effective meta-algorithm, and the integration of MO-CMA-ES can be a good future work.
>
> Note that the key is not whether a model has been built or not for the function to be optimized (or so-called model-based vs model-free from the reviewer's point of view), but whether the built model is efficient in capturing the essence of the function. In our work, we choose to build classifiers to quickly get rid of bad regions and only build local models to focus on important regions. This is typically more efficient than a global GP model for the entire search space.
>
> ---
> **8. MOBO Performance**
>
> We provided the experiment setting in **appendix.G** and note that we decrease the raw sample(Daulton et al. (2020)) from 1024 to 256 to match the sample budget of all methods for comparison, which speeds up the search but may lead to lower performance. However, LAMOO+qEHVI with 256 raw samples still have better results than the original qEHVI results reported in Daulton et al. (2020).

---

> ### Author Response · Authors · 2021-11-19
> **Response to Reviewer 2MfT (3/3)**
>
> ---
> **9.Wall Clock Run Time**
>
> We reported the wall clock run time in **appendix.E**.
>
> We would like to stress that for qEHVI, LAMOO as a meta-algorithm slightly accelerates the original algorithm, that is because by reducing the size of search space, the optimization of the acquisition function is easier. However, LAMOO plugging in CMA-ES costs significantly higher times compared to its vanilla version. That is because we searched 1000 points for CMA-ES, with an increasing number of samples generated, the training times of SVM in each node are increased accordingly. However, in many real-world tasks, the most time-consuming part of the search is the evaluation time of the searched time (e.g. Neural architecture search[r-4][4-5]), in which one function evaluation can take hours or days. Therefore, we view sample efficiency as more important in practice.
>
> ---
> **Minor Issues**
>
> Thank you for your suggestion. We have changed the citations of multiple works in chronological order.
>
> ---
> **Reference**
>
> [r-1] Q. Xu, Z. Xu and T. Ma, "A Survey of Multiobjective Evolutionary Algorithms Based on Decomposition: Variants, Challenges and Future Directions," in IEEE Access, vol. 8, pp. 41588-41614, 2020, doi: 10.1109/ACCESS.2020.2973670.
>
> [r-2] R. C. Purshouse and P. J. Fleming, "Evolutionary many-objective optimisation: an exploratory analysis," The 2003 Congress on Evolutionary Computation, 2003. CEC '03., 2003, pp. 2066-2073 Vol.3, doi: 10.1109/CEC.2003.1299927.
>
> [r-3] Hisao Ishibuchi, Takashi Matsumoto, Naoki Masuyama, and Yusuke Nojima. 2020. Effects of dominance resistant solutions on the performance of evolutionary multi-objective and many-objective algorithms. In Proceedings of the 2020 Genetic and Evolutionary Computation Conference (GECCO '20). Association for Computing Machinery, New York, NY, USA, 507–515. DOI:https://doi.org/10.1145/3377930.3390166
>
> [r-4] Zoph, Barret, et al. "Learning transferable architectures for scalable image recognition." Proceedings of the IEEE conference on computer vision and pattern recognition. 2018.
>
> [r-5] Real, Esteban, et al. "Regularized evolution for image classifier architecture search." Proceedings of the aaai conference on artificial intelligence. Vol. 33. No. 01. 2019.

---

> > ### Comment · Reviewer_2MfT · 2021-11-23
> > **Follow-up Comments**
> >
> > Thank you for your detailed response. Many of my concerns have been properly addressed, so I raise my score to 6. Here are my follow-up comments and concerns:
> >
> > **1. Novelty and Contribution**
> >
> > In my understanding, the crucial idea ("learn to partition") along this research line was first proposed and significantly developed in LaNAS and LaMCTS. While LaP^3 and LaMOO are two reasonable extensions (e.g., LaMCTS + X) to tackle planning and multi-objective optimization problems, respectively.
> >
> > I do aware of the decades-long research on MOO. However, the idea to use dominance relationship or indicator (e.g., hypervolume) to generalize SOO to its MOO counterpart in many different ways for different algorithms is now well-known and widely used in the MOO community. The example of CMA-ES v.s. MO-CMA-ES is one of the early attempts that was proposed 15 years ago. In addition, different classification/partition based methods have been proposed for SOO and MOO as listed in the related work [1-9] (which were not included in the original submission).
> >
> > That is why I think the proposed LaMOO method is a "novel", "reasonable", "direct" and "incremental" generalization from single-objective LaMCTS to LaMOO. Indeed I have listed the generalization as one of the strengths for this work, with the concerns in the weaknesses. It is a bit unfair to claim I have said LaMOO is "simple" or even "trivial" (which I never said) and use the term "awkward".
> >
> > **Related Work**
> >
> > Thanks for the proper discussion. (Some of) these related works might deserve to be discussed in the main paper.
> >
> > **2/3. Theoretical Analysis and Exploitation v.s. Exploration**
> >
> > Thanks for the clarification and the discussion on setting C_p. Is it possible to consider the exploitation v.s. exploration trade-off in the theoretical analysis? It seems that the exploration ability with MCTS is the crucial component that distinguishes LaMCTS from other classification-based methods (e.g., [1-3]).
> >
> > **7/9. "Model-Free Algorithm", Time Complexity, and Wall Clock Run Time**
> >
> > Thank you for the clarification. I totally agree with the authors that the evaluation for real-world optimization problems could be extremely time-consuming (e.g., it can take hours or days), so the sampling efficiency (e.g., number of evaluations) is more important in practice. In this case, the time complexity of LaMOO could be trivial compared with the expensive evaluation. This is also the default assumption for Bayesian optimization.
> >
> > But, what is the scenario we should actually need LaMOO + MO-CMA-ES (or NSGA-III)?
> >
> > If sampling efficiency is the crucial concern, the original qEHVI and LaMOO + qEHVI are much better than LaMOO + MO-CMA-ES. That is why I think global GP models could be more efficient than the classification model in LaMOO in this case (though we can combine them as in LaMOO + qEHVI). If the actual wall clock time is crucial (with cheap evaluations), the vanilla MO-CMA-ES and NSGA-III would be much faster than the LaMOO+ version with high time complexity. It seems LaMOO + MO-CMA-ES could be suitable for the application with "not so expensive while not so cheap evaluation" which could be vague to identify in practice.
> >
> > **3 - 6, 8:**
> > Thank you for the new experiments, analyses, and discussions.

---

> > > ### Author Response · Authors · 2021-11-24
> > > **Response to the follow-up questions**
> > >
> > > Thanks for the insightful comments.
> > >
> > > **1. Novelty**
> > >
> > > We totally agree that the idea of learning-to-optimize has been leveraged in the previous works (e.g., LaMCTS, LaNAS, LaP^3) for single-objective optimization, and in LaMOO, we leverage this idea for multi-objective optimization. We have acknowledged it in our current version of the paper. We will add more explanations to clarify our contribution in the next revision.
> > >
> > > We are sorry to use the word “awkward” in our rebuttal. We now understand your motivation to reference CMA-ES and MO-CMA-ES separately. Thanks for the clarification.
> > >
> > > ---
> > >
> > > **2. Related work**
> > >
> > > We will incorporate them in the main text in the next revision.
> > >
> > > ---
> > >
> > > **3.Theoretical analysis**
> > >
> > > We have to admit that it is quite non-trivial to incorporate the exploration-exploitation trade-off into our analysis. This key difficulty lies in how to model the process of re-picking the optimal solution with the help of exploration after the initial bad assignment, and how such a re-picking changes the partition learning and future exploration.
> > >
> > > With MCTS, such a learning-to-optimize algorithm will eventually pick the correct solution, since each leaf will be visited infinitely many times. The key is its sample complexity. The intuition here is that as long as the optimal solution is being visited once, it will be “dragged” to the updated good partition once the partition is re-learned with new data points. Therefore the sample complexity is likely to be low.
> > >
> > > While we will continue working on this part, due to its technical complexity, we may likely leave it to future work.
> > >
> > > ---
> > > **4. "Model-Free Algorithm", Time Complexity, and Wall Clock Run Time**
> > >
> > > We kindly disagree that a global GP model is able to model the entire landscape of the function well if sample complexity is the critical part to consider. While a global GP model carries the promise to be adaptive to any function, it also needs to model every corner of the function. If initially, GP is too optimistic in one region but too pessimistic in other regions, then it may trap the search to local minima. The same situation may also happen for EA, which lacks a global mechanism to explore.
> > >
> > > On the other hand, classification-based methods may avoid modeling a region with a complicated landscape altogether and leave its job to local models that focus on local regions. MCTS could ensure bad regions are still frequently explored in case there are any good solutions hiding. Therefore, LaMOO may have an advantage.

---

> > > > ### Comment · Reviewer_2MfT · 2021-11-29
> > > > **Thank you for the further response**
> > > >
> > > > Thank you for the further response. I keep the positive score to this work.

---

### Official Review · Reviewer_xfNd · 2021-11-02

**Correctness:** 4
**Technical Novelty And Significance:** 3
**Empirical Novelty And Significance:** 3
**Recommendation:** 8
**Confidence:** 4

**Main Review:**

The paper is well-written and, in my view, it offers a simple but effective "meta" perspective for multiobjective programming. In particular, it resembles many criterion-search space techniques that split the objective space for strategic exploration. However, here authors use the dominance metric to partition the primal space, which leads to interesting properties and a tree encoding.

I also appreciate the theoretical aspects investigated by the authors, which are sound to the best of my analysis and give a more concrete justification to the results we observe in the numerical setting.

My major concern, however, is that it is somewhat unclear when the proposed meta-algorithm is applicable or more beneficial. In particular:

(a) Are there (combinatorial) problems where the tree would need to be significantly large to ensure high-quality points are actually found? I was somewhat confused as to how the tree may grow depending on the underlying structure of the problem.

(b) What are the properties that a multiobjective algorithm would need to satisfy to apply the split partition? For example, suppose one is using a multiobjective mixed-integer linear program to enumerate the frontier. Is it possible to represent complex splits in a way that they encode true partitions? It seems that broader split definitions, for instance in larger-dimension criteria space, could be high non-linear (or perhaps I misunderstood some concept?).

(c) What are the guidelines to set the exploration factor C_p empirically? It seems that it is highly optimized for the numerical experiments that were performed in the paper

**Summary Of The Paper:**

The paper develops an enhancement to multiobjective solvers so to find better Pareto solutions. The idea is to learn a proxy of the distance of samples to the Pareto frontier, and leverage such information to split the search space via a tree structure, Samples can then be extracted from promising nodes of the tree from other multiobjective algorithms. Numerical results investigate the performance of the method for both synthetic and practical benchmarks.



**Summary Of The Review:**

A well-written paper that provides an interesting and relevant contribution to multiobjective optimization. One of the key benefits of the method is its generality, in that it can be incorporated into existing multiobjective solvers, and simple implementation. Numerical results suggest significant improvements with little implementation work.

---

> ### Author Response · Authors · 2021-11-19
> **Response to Reviewer xfNd**
>
> We sincerely thank the reviewer for your insightful comments and suggestions, please check our answers below.
>
> ---
> **Are there (combinatorial) problems where the tree would need to be significantly large to ensure high-quality points are actually found?**
>
> We acknowledge that there exist hard pathological combinatorial problems where the tree size needs to be exponential in order to ensure high-quality points to be found in each leaf. For example, consider the function f = $x_{1}$ xor $x_{2}$ xor $x_{3}$… xor $x_{n}$ + eps * I($x_{1}$ =1, $x_{2}$ =0, …, $x_{n}$ =1). In this case, SVM classifiers with smooth decision boundaries won’t be able to distinguish good from bad regions, until we cut the regions into very fine-grained levels. In this case, the number of tree nodes has to be exponential to really find the right corner that contains the optimal solution.
>
> However, in general when the optimization problem has regularity (e.g., most of the regions are not good at all), then LaMOO would work well. This is characterized by our theoretical analysis, as well as empirically verified in our experiment section.
>
> ---
> **What are the properties that a multiobjective algorithm would need to satisfy to apply the split partition?**
>
> As shown in our **Observation 1 & 2 (Sec. 3.3)**, in the multi-objective optimization, if the Pareto optimal set lies in a small region in the search space, and/or the optimal set can be well-separated from non-dominant regions, then learning a partition definitely helps remove bad regions and improves the sample efficiency.
>
> Whether the complex splits can be represented heavily depends on whether its boundary can be learned efficiently and effectively given the previous observations of the function. For example, if we have prior knowledge that good/bad regions are split by a sinusoidal decision boundary and we fit it with a sinusoidal function, then with very few samples, the boundary can be well predicted.
>
> Overall, the reviewer exactly demonstrates the situation where the traditional (not learning-based) space partition approach fails: the required splits are too complicated and cannot be characterized by simple constraints (like linear or quadratic). In such situations, our approach can be advantageous since its partition is learned from previous data points and can be adaptive to the specific properties of multi-objective functions.
>
> ---
> **What are the guidelines to set the exploration factor C_p empirically?**
>
> Please check the answers to common questions(**Guidelines to set the exploration factor C_p**). Thanks.

---

> > ### Comment · Reviewer_xfNd · 2021-11-22
> > **Acknowledgement**
> >
> > Thank you for addressing my questions, I appreciate it.

---

### Official Review · Reviewer_vdwH · 2021-11-08

**Correctness:** 4
**Technical Novelty And Significance:** 3
**Empirical Novelty And Significance:** 4
**Recommendation:** 8
**Confidence:** 4

**Details Of Ethics Concerns:**

I did not find any ethical concerns.

**Main Review:**

Strengths.
- The proposed method uses an interesting combination of algorithms.
- The results are compelling.
- The paper is well written. It is easy to understand, well-organized, and the illustrations help understand the behavior of the algorithm.

Weaknesses.
- It is possible to criticize that it is a combination of existing techniques
(which is not a serious issue, in my opinion).
- Some papers propose Multi-Objective optimization using A* search.
I think the related work section should mention these.
The following short paper covers some of such papers as of 2011.
https://www.ijcai.org/Proceedings/11/Papers/483.pdf
- A part of the algorithm that "updating all previous samples" may take a long time for larger-scale problems because the time will be O(N^2) to the sample size N.

**Summary Of The Paper:**

This paper proposes to use MCTS to enhance existing algorithms for Multi-Objective Optimization problems.
Experimental results on a wide range of benchmark problems show the improving performance of the proposed method, LaMOO.


**Summary Of The Review:**

In my opinion, the significant improvement in performance with a simple method is highly valuable.
The idea is simple, well-explained, and the results are promising.
This paper should be valuable for many applications.

---

> ### Author Response · Authors · 2021-11-19
> **Response to Reviewer vdwH**
>
> We sincerely thank the reviewer for your insightful comments, please check our answers below.
>
> ---
> **Add A\* search to related works**
>
> Thanks for pointing out the missing related works. We added them in our revision.
>
> ---
> **O(N^2) time complexity to update all samples**
>
> We only consider updating all samples in a selected leaf. The number of samples(N) in a leaf is not very large(typically from 10 - 30).

---

> > ### Comment · Reviewer_vdwH · 2021-11-29
> > **Response to the response**
> >
> > Thank you for the reply. It resolved my concerns.

---

### Author Response · Authors · 2021-11-19
**Answer to the common questions from all reviewers**

We are glad to see most of the reviewers are positive with the paper and sincerely thank reviewers for their insightful comments, please check our answers to the common questions below.  We also colored the new contents in the paper to purple.

---
**Computational complexity & computational overhead of LaMOO**

Thanks to the reviewers for raising this important question.

First, we want to emphasize that in real-world applications in which it takes hours, days, or even weeks to evaluate one data point to retrieve the values of M objectives, the overhead given limited samples becomes less important. That’s why we choose to use sample efficiency to evaluate our proposed algorithm.

In our algorithm (**Alg. 1**), in each node, the hypervolume over all samples need to be computed, which is O(n log n) for M <= 3 (See [r-2]) and O(n^(M/2) + n log n) if M > 3 (See [r-1]), where n is the number of samples in each node and M is the number of objectives. At the root node, n can be as large as the total number of samples. Note that it takes O(n^2) to train a nonlinear SVM classifier in each node, therefore HV computation is likely to be the dominating factor when there are many objectives.

The detailed breakdown of the computational complexity is now listed in Appendix I.

That being said, while the computational complexity is high, in practice, LaMOO+qEHVI is often faster than qEHVI as shown in **figure.10** in the **appendix.E**. We suspect that these qEHVI multi-objective optimizers also suffer similar issues.

**A cheaper variation with comparable performance**. Furthermore, we also present a variation of our Alg. 1, with similar sample complexity to achieve the same HV but much cheaper computational cost. In **appendix.J**, we have already shown that it has similar performance as Alg. 1 given the same function evaluations, in small-scale problems(see figure.15 and figure.16), and we are currently evaluating it on large-scale problems. The intuition here is to directly search over the leaves of the tree and pick the one with the best UCB score. Then we avoid HV computations at intermediate nodes and only need to compute HV at each leaf node, which typically contains much fewer samples (e.g., 10-30) than the root node. Still, nonlinear SVM classifiers need to be trained at each intermediate node, yielding quadratic computational complexity.

Note that computing hypervolume is a hard problem (#P-hard, see [here](https://arxiv.org/pdf/0812.2636.pdf)) and all multiple-objective optimization techniques that compute hypervolume algorithmically would meet with the same issue. To improve on HV computation is beyond the scope of this paper. We acknowledge this is a limitation and will mention it explicitly in the next revision.

---
**Guidelines to set the exploration factor C_p**

As we mentioned in **Sec.5.3**, a "rule of thumb" is to set the Cp to be roughly 10% of the maximum hypervolume HVmax. If HVmax is $\textbf{unknown}$, Cp can be dynamically set to 10% of the hypervolume of current samples in each search iteration. The **figure.9** in **Appendix.D** demonstrates the difference between 10% HVmax and 10 % HVcur in three problems(Branin-Currin, VehicleSafety, and  Nasbench201). The final performances by using 10% HVmax and 10 % HVcur are similar.

---
**Reference**

[r-1]Nicola Beume. 2009. S-metric calculation by considering dominated hypervolume as klee's measure problem. Evol. Comput. 17, 4 (Winter 2009), 477–492. DOI:https://doi.org/10.1162/evco.2009.17.4.17402

[r-2]N. Beume, C. M. Fonseca, M. Lopez-Ibanez, L. Paquete and J. Vahrenhold, "On the Complexity of Computing the Hypervolume Indicator," in IEEE Transactions on Evolutionary Computation, vol. 13, no. 5, pp. 1075-1082, Oct. 2009, doi: 10.1109/TEVC.2009.2015575.

---

### Decision · Program_Chairs · 2022-01-20

**Decision:**

Accept (Poster)

**Comment:**

Multi-objective learning is an increasingly important topic. This paper presents a method for better finding parts of the Pareto frontier through a new method to estimate the distance to the frontier and use this proxy to refine the state space partition.  The reviewers found this paper interesting and compelling and generally well written. The reviewers also thought the work could be further improved by better clarifying in the text where the proposed approach might fail, and what properties of the domain are needed, and also to better situate this paper within the related work, potentially including additional experimental comparisons. The authors provided detailed responses to the proposed questions and the authors are encouraged to ensure that these suggestions and discussions are well represented in the revised version.